# The Role of NMDA Receptor Partial Antagonist, Carbamathione, as a Therapeutic Agent for Transient Global Ischemia

**DOI:** 10.3390/biomedicines11071885

**Published:** 2023-07-03

**Authors:** Jigar Pravinchandra Modi, Wen Shen, Janet Menzie-Suderam, Hongyuan Xu, Chun-Hua Lin, Rui Tao, Howard M. Prentice, John Schloss, Jang-Yen Wu

**Affiliations:** 1Department of Biomedical Sciences, Charles E. Schmidt College of Medicine, Florida Atlantic University, Boca Raton, FL 33431, USA; 2Center of Complex Systems and Brain Sciences, Florida Atlantic University, Boca Raton, FL 33431, USA; 3Program in Integrative Biology, Florida Atlantic University, Boca Raton, FL 33431, USA; 4Department of Nursing, Kang-Ning University, Taipei 11485, Taiwan; 5Department of Pharmaceutical Science, American University of Health Sciences, Signal Hill, CA 90755, USA

**Keywords:** Carbamathione (Carb), glutamate, hypoxia, endoplasmic reticulum (ER) stress, PC-12 cell culture, stroke, whole-cell patch clamp and bilateral carotid artery occlusion (BCAO)

## Abstract

Carbamathione (Carb), an NMDA glutamate receptor partial antagonist, has potent neuroprotective functions against hypoxia- or ischemia-induced neuronal injury in cell- or animal-based stroke models. We used PC-12 cell cultures as a cell-based model and bilateral carotid artery occlusion (BCAO) for stroke. Whole-cell patch clamp recording in the mouse retinal ganglion cells was performed. Key proteins involved in apoptosis, endoplasmic reticulum (ER) stress, and heat shock proteins were analyzed using immunoblotting. Carb is effective in protecting PC12 cells against glutamate- or hypoxia-induced cell injury. Electrophysiological results show that Carb attenuates NMDA-mediated glutamate currents in the retinal ganglion cells, which results in activation of the AKT signaling pathway and increased expression of pro-cell survival biomarkers, e.g., Hsp 27, P-AKT, and Bcl2 and decreased expression of pro-cell death markers, e.g., Beclin 1, Bax, and Cleaved caspase 3, and ER stress markers, e.g., CHOP, IRE1, XBP1, ATF 4, and eIF2α. Using the BCAO animal stroke model, we found that Carb reduced the brain infarct volume and decreased levels of ER stress markers, GRP 78, CHOP, and at the behavioral level, e.g., a decrease in asymmetric turns and an increase in locomotor activity. These findings for Carb provide promising and rational strategies for stroke therapy.

## 1. Introduction

In the US, the third leading cause of death and disability is stroke, as reflected in the fact that one person dies every 4 min from a stroke [1]. Stroke diminishes mobility in more than half of stroke survivors aged 65 and over. Post-stroke, a clinical trial indicated that more than 60% of patients suffer from cerebral hypoxia in the first 60 h [2]. A rise in the extracellular concentrations of excitatory amino acids and glutamate is caused by stroke or ischemia [3,4]. This rise in glutamate could be related to an increasing release from neurons or derived from energy failure or due to lowered clearance of glutamate by glial transporters. A low amount of glutamate is crucial for neuronal function compared to a high amount of glutamate, which elicits cell death and is excitotoxic [5]. High levels of glutamate stimulate N-methyl-D-aspartate (NMDA) receptors and the process of calcium-dependent neurotoxicity, which is the primary reason for the death of neurons in hypoxic–ischemic brain injury [6].

The endoplasmic reticulum (ER) is the location where newly synthesized proteins are folded and processed. There is an increasing indication that ER stress plays a fundamental role in hypoxia/ischemia-induced cell dysfunction [7,8,9]. Hypoxia initiates the accumulation of unfolded proteins in the ER, which leads to the unfolded protein response (UPR) and ER-associated protein degradation (ERAD) [10]. ER functions are restored by UPR and initiated in mammalian cells by the activation of three distinct types of stress sensors located on the ER membrane, namely double-stranded RNA-activated protein kinase-like ER kinase (PERK), activating transcription factor 6 (ATF6), and inositol-requiring kinase 1a (IRE1a) [11]. Neuronal homeostasis and the levels of glucose-regulated protein 78 (GRP 78), an ER chaperone, contribute to the activation of PERK, ATF6, and IRE1 [3]. In ER dysfunction, detachment of GRP 78 from the stress sensor leads to the dimerization and phosphorylation of PERK and IRE1 and to the cleavage of ATF6 (P90) into ATF6 (P50). PERK activation initiates phosphorylation of eukaryotic initiation factor 2a (eIF2a), resulting in a general attenuation of protein translation, which is one mechanism that prevents the overload of proteins in the ER. eIF2a phosphorylation permits specific translation of activating transcription factor 4 (ATF4), which upregulates many important genes that function in redox control, metabolism, and protein folding under persistent or severe ER stress [6]. Upon activation, IRE1a functions as a serine–threonine kinase and endoribonuclease, which initiates the unconventional splicing of mRNA for transcription factor Xbox binding protein-1 (XBP1).

Activation of the three ER stress pathways initiates up-regulation of the transcription factor C/EBP homologous protein (CHOP)/GADD153 (growth arrest and DNA damage-inducible gene 153) [12]. All downstream signals, such as the activation of transcription factors and kinase pathways and the regulation of BCL2 family members, ultimately elicit caspase activation, resulting in the ordered and sequential dismantling of the cell [13]. Autophagy contributes to preserving intracellular homeostasis. An increase in Beclin 1 (an important initiator of autophagy) is demonstrated in both acute and severe ischemia, which causes excessive autophagy [14]. Caspase-12 is known to be a key contributor to apoptosis that is induced by ER-stress [15].

Disulfiram is a drug that has been used for over 65 years for the treatment of alcohol abuse [16], and Carbamathione (Carb) is a metabolite of disulfiram [17]. Disulfiram needs to be activated into S-methyl N, N-diethylthiolcarbamate sulfoxide (DETC-MeSO) for its inhibitory action on ALDH2 [18,19]. In vivo, DETC-MeSO is further oxidized to the sulfone form, which is carbamoylated to form Carb, a glutathione adduct. Carb is a partial NMDA glutamate antagonist and does not inhibit ALDH2 [17,20]. In cocaine dependence, Carb proved to be useful as a pharmacological agent with the advantage that it lacks ALDH2 inhibitory activity [20]. Faiman et al., 2013 showed that the IV administration of Carb to mice irreversibly blocks glutamate binding to mouse brain synaptic membranes [17], and they proposed that Carb may affect both the N-methyl-D-aspartate (NMDA) and non-NMDA glutamate receptor subtypes. The brain dopamine levels are increased by NMDA receptor antagonists [21]. In a dose-dependent manner, the increasing of Carb concentrations (20, 50, 200 mg/kg iv) increased DA, decreased GABA, and had a biphasic effect on glutamate, first increasing and then decreasing glutamate in both the NAc and mPFC [22].

In total, the mechanisms underlying the functions of Carb are still not fully understood. Our previous studies demonstrated the protective effect of DETC-MeSO on primary neuronal cultures in hypoxia/reoxygenation conditions and in an animal model of stroke [6]. The hypothesis of this study is that “Carb provides neuroprotection via inhibiting apoptosis and ER stress in brain in an animal model of stroke”. Therefore, the aim of this study is to investigate the protective effect of Carb against ER stress apoptosis in (1) a stroke-like cell culture model involving glutamate-induced or hypoxia/reoxygenation-induced cell death and (2) a mouse model of global cerebral ischemia, namely, the bilateral carotid artery occlusion (BCAO) model. 

PC12 cells represent a commonly established neuronal model system. This cell model is also generally used to study cellular glutamate toxicity [23]. PC12 cells are sensitive to glutamate or hypoxia/reoxygenation injury; therefore, we proposed that it is appropriate to use PC12 cells to examine whether Carb offers protection against glutamate or hypoxia-induced cytotoxicity.

We carry out whole-cell patch clamp recordings of glutamate receptor currents in mouse retinal ganglion cells in a whole-mounted retinal tissue preparation to verify the effects of Carb against glutamate receptor-mediated toxicity in the CNS.

## 2. Materials and Methods

### 2.1. Materials

We purchased F-12K media, trypsin-EDTA solution, horse serum, fetal bovine serum, and the rat pheochromocytoma (PC) PC-12 cell line from ATCC (Manassas, VA, USA). Poly-D-lysine was acquired from Sigma (St. Louis, MO, USA). An Adenosine 5′-triphosphate (ATP) Bioluminescent Assay Kit was bought from Promega (Madison, WI, USA) (Appendix A).

### 2.2. In Vitro Study

The in vitro study was designed to establish the efficacy of Carb in exerting a protective function against cell death. Glutamate was used to mimic glutamate-induced toxicity in the rat pheochromocytoma (PC12 cell) line, and in order to simulate ischemia-like conditions in the PC-12 cells, hypoxia/reoxygenation was used. Then, the effect of Carb against glutamate-induced toxicity (First phase) or hypoxia/reoxygenation injury (Second phase) was determined by calculating cell viability.

#### 2.2.1. PC-12 Cell Line Culture

In a Petri dish, PC-12 cells were cultured for one week prior to use. PC-12 cells were maintained in an incubator at 37 °C/5% CO_2_, and cells were fed every other day using F12-K medium supplemented with 5% (*v*/*v*) fetal bovine serum (FBS), 10% (*v*/*v*) heat-inactivated horse serum (HS), and 1% (*v*/*v*) penicillin–streptomycin solution. All experiments were conducted on undifferentiated cells plated in 96-well plates at a density of approximately 5 × 10^4^ cells/mL for the ATP assay [24]. On the first day of use, cells were harvested by first adding 2 mL trypsin and then incubated for 15 min. Next, 2 mL of fresh medium was used as a wash, and the cells were centrifuged and re-suspended. Cell density was measured by cell counting using a hemocytometer and a tissue culture microscope. After cell density was determined, 96-well plates and 6-well plates were plated with 2.5 × 10^4^ cells per well and 5 × 10^5^ cells per well for the Adenosine 5′-triphosphate (ATP) assay and Western blot analysis, respectively. Cells were incubated for 24 h, and varying Carbamathione concentrations were added along with 10 mM glutamate/hypoxia at 24 h and reoxygenation at 24 h.

Cell lines:

The cell line was purchased from ATCC (cat # CRL-1721.1) and was last validated and certified in 2018. 

The cell line is used up to a maximum number of five passages.

#### 2.2.2. Glutamate Toxicity

Cell cultures were pre-incubated with 0.5 µM to 500 µM concentrations of Carb for 1 h. Then, to induce glutamate toxicity, the neurons were treated with glutamate (10 mM) for 24 h [11,24].

#### 2.2.3. Hypoxia/Reoxygenation

To simulate ischemic conditions, PC-12 cells in 6- or 96-well plates were placed in a hypoxia chamber with oxygen levels regulated at 0.3–0.4% [6,24]. The level of oxygen was regularly supervised using an oxygen electrode. PC-12 cell cultures in the absence or presence of Carb were exposed to 24 h of hypoxia. Reoxygenation was achieved by shifting cultured plates from the hypoxic chamber and transporting them into a normal culture incubator where they remained for another 24 h.

#### 2.2.4. Carb Concentration

Carb 0.15 M was prepared in 0.15 M sodium bicarbonate solution as described in Kaul et al. [20].

#### 2.2.5. Measurement of Cell Viability Using the ATP Assay

PC-12 cells in 96-well plates were treated with or without Carb for 1 h, and then the cells were exposed to glutamate/hypoxia–re-oxygenation conditions to induce cell death. ATP solution was added to each well, and the cells were incubated for 10 min, after which the amount of ATP was measured using a luciferase reaction. The luminescence intensity was finalized using a luminometer with lysates in a standard opaque-walled, multi-well plate. The ATP content was regulated by running an internal standard and stated as a percentage of untreated cells (control) [25].

#### 2.2.6. Electrophysiological Recording

Whole-cell patch-clamp recordings were performed on ganglion cells in dark-adapted, flat-mounted mouse retinas using an EPC-10 signal amplifier and HEKA Pulse software Version: 1.3 (HEKA Instruments Inc., Lambrecht/Pfalz, Germany) [26]. The recording electrodes were filled with a high-potassium solution containing (in mM): potassium gluconate (100), KCl (5), MgCl2 (1), EGTA (5), HEPES (10) with an ‘ATP regenerating cocktail’ consisting of (in mM) ATP (20), phosphocreatine (40), and creatine phosphatase (2); pH 7.2. Glutamate synaptic currents were elicited using a photic stimulus delivered with a green light-emitting diode (LED; peak emission of 520 nm) focused directly on the flat-mount retinal tissues and controlled with the output of the HEKA amplifier. The intensity of the stimulus was 1.6 lux, which evoked both rod and cone activity. A gravity-driven system was used for bath perfusion and manually controlled for delivering drugs during the experiments.

### 2.3. In Vivo Study

The in vivo study was designed to establish the efficacy of Carb in exerting a protective function in an animal model of stroke.

#### 2.3.1. Animal Preparation

Vertebrate Animal Study: ‘Florida Atlantic University Division of research, Institutional Animal Care and Use Committee provided full approval for this research’, ‘A16-08/A18-29/A19-14’ and ‘A-3883-01/D16-00507’. Note: the protocols were A16-08; A18-29 and A19-14. FAU Animal Welfare Assurance Number D16-00507.

Male Swiss Webster mice (20 weeks of age) were obtained from Charles River laboratory. The rationale for using young adult mice was to minimize the complications of metabolic changes associated with adult animals. It would be desirable to include aged mice in future investigations. Males were used in this study so that it would be possible to compare our levels of protection as well as protective mechanisms with those observed previously for DETC-MeSO, which examined neuroprotection in males [6]. Animals were fasted overnight before surgery but were allowed access to water only. Ketamine (100 mg/kg, i.p.) plus xylazine (10 mg/kg, i.p.) was used to anesthetize the mice. A gaseous mixture consisting of 30% oxygen, 70% N_2_O, and 0.5% isoflurane was used to induce anesthesia in the mice, and to maintain anesthesia, 0.5% isoflurane was used. The mice breathed spontaneously with a breathing mask throughout the surgical procedure. A rectal temperature probe was inserted. The mice were kept on a thermostat-controlled heating pad, maintaining a constant core temperature of 37.0 ± 0.5 °C during the surgical procedure.

#### 2.3.2. Animal Care, Feeding, Housing, Monitoring, and Enrichment

The Animal Use Program at Florida Atlantic University is accredited by AAALAC, which is a clear indication of the exceptionally high level of standards that are maintained at the University for animal care and use. The study animals were imported from the supplier, and once in-house, they were checked daily for health status. The animals acclimated for 3 days after import from the supplier. The animals were housed in group housing conditions with environmental enrichment and free access to food and water. 

Animal care and housing were both of very high quality and were overseen by the Office of Comparative Medicine. Feeding involved standard mouse chow, to which the animals had access ad libitum. Environmental enrichment was provided for all animals, and the type of enrichment provided was determined in conjunction with Comparative Medicine and was approved by the IACUC. The animal/cage locations were randomized. 

#### 2.3.3. Bilateral Common Carotid Artery Occlusion (BCAO)

After anesthesia, each mouse was placed in a supine position. Adhesive tape was used to hold the animal’s tail and paws to the heating pad. A sagittal ventral midline incision (~1 cm length) was performed. To visualize the underlying common carotid artery (CCA), salivary glands were carefully separated and mobilized on each side. Both CCAs were cautiously detached from the vagal nerves and accompanying veins without injuring these structures. It is critical to avoid any handling of the vagal nerves, which could damage the parasympathetic nervous system and induce major cardiac arrhythmia or even irreversible cardiac arrest. After isolating both common carotid arteries (CCAs), they were occluded using non-traumatic aneurysm clips. An operating surgical microscope was used to confirm the complete interruption of blood flow. The aneurysm clips were removed from both CCAs after 30 min of ischemia. Restoration of blood flow (reoxygenation) was seen directly under the microscope. Sham-operated controls were performed with the same surgical procedures except that CCAs were not occluded [27]. Body temperature was observed and maintained at 37 °C ± 0.5 °C during surgery and during the immediate postoperative period until the animals recovered fully from anesthesia.

#### 2.3.4. Corner Tests

When animals encounter a corner, they turn asymmetrically due to brain injury which is determined using the corner test. An experimental 30° corner was setup using two boards (with dimensions of 30 × 20 × 1 cm^3^) in such a way that a small opening was left along the joint between the boards. A mouse was placed at a 12 cm distance from the corner and allowed to walk into corner so that the vibrissae on both sides of the animal’s face contacted the two boards simultaneously. We conducted behavior tests (stratification) on all mice to screen for mice with no turning asymmetry (*n* ≥ 18) before the BCAO procedure. We calculated the percentage of turns to the right or left side of each mouse, recording only those turns involving full rearing along one of the boards [27]. This stratification procedure excluded mice with 80–100% asymmetric turns (*n* = 4); we included mice that turned in either direction (*n* = 14) with a pretest score of 0.50 ± 0.08. Each mouse took part in ten trials for up to 4 days after BCAO.

#### 2.3.5. Locomotion (Force-Plate Actometer) Test

The force-plate actometer is a combination of mechanical, electronic, and computing basics that represent mathematical and physical principles used to quantify whole-organism behavioral traits with application to basic neuroscience research. The methods for calibration and details of data acquisition are described elsewhere [28,29]. Briefly, the force-plate actometer bought from BASi Corp (model FPA-I; West Lafayette, IN, USA) consists of a force-sensitive plate at a resolution of 200 Hz, with a sound attenuation chamber, a computerized data acquisition board, and analysis system software (FPA 1.10.01). To measure locomotor activity, a newly developed force-plate actometer was utilized. Each animal was placed on the force-plate actometer for one separate 60 min session. Locomotor activity of BCAO mice with/without Carb treatment was tested after 4 days. Between each test, the plate was thoroughly cleaned with paper towels followed by a deodorant treatment (70% ethanol, 1% acetic acid, and then water) to remove animal waste (i.e., feces, urine, saliva, and fur) and odor. Trace data on movements were automatically stored in the hard drive for offline analysis, and locomotion changes were determined using power spectral analysis, which was expressed as an arbitrary distance. The unit for changes in the power force was arbitrary.

#### 2.3.6. Mouse Groups and Treatment Schedules

The animals were randomly assigned to sham, control, and experimental groups (Total N = 27). Thereafter, in the experimental group (Carb treated group, *n* = 9), Carb (14 mg/kg in 0.3 mL saline 0.9%) was injected subcutaneously 30 min after occlusion and continued daily until the animal was sacrificed. In the control group (vehicle-treated group, *n* = 9), vehicle (0.3 mL saline 0.9%) was injected subcutaneously 30 min after occlusion. Each mouse received vehicle for 4 days before sacrifice. The sham-operated group (*n* = 9) received the same surgical procedure without occlusion of CCA. The animals were allowed to recover from anesthesia after surgery and were given food and water ad libitum. The animals were observed daily for body temperature and weight, and those who had body temperature more than 39 °C after 24 h were omitted from the experiment [30].

#### 2.3.7. Details of Euthanasia Method(s) Used

The euthanasia method used in this study was Pentobarbital administration (IP).

#### 2.3.8. Criteria Established for Euthanizing Animals Prior to the Planned End of the Experiment and Whether This Was Needed

If an animal showed signs of stress or illness after the surgery, it was examined by Veterinary services. These experiments used a 45 min BCAO method (a shorter duration of occlusion than that used in some previously published studies). With this 45 min period of BCAO, the surgery did not result in unanticipated mortalities over the duration of the project. Pain associated with surgery was alleviated with anesthetics and analgesics (Buprenex-slow release or Buprenorphine). The PI, project personnel, and Veterinary Services observed and monitored animals for signs of pain and distress.

Adverse events reporting: Because large mice (Swiss Webster body weight 36 g to 44 g) were used, and because of the short occlusion time (30 min), an estimated mortality rate of 20% was anticipated in contrast to the published BCAO mortality rate of close to 70%. There were no unexpected adverse events, and survival was >90%. A low incidence of mortality was observed in the control stroke animals that received no therapy.

Endpoints in this study: Humane endpoints that were monitored included poor posture, loss of body weight, and delayed wound healing. These observed changes prompted consultation with veterinary services.

Animals were excluded if their health condition deteriorated to the level of humane endpoints, at which time they were examined by Veterinary Services. Because there were very few mortalities, the BCAO procedure with 30 min occlusion was found to be less severe than the procedures reported in previous publications that involved smaller mice and a longer duration (60 min) of occlusion.

#### 2.3.9. Assessment of Lesion Size Using 2,3,5-Triphenyltetrazolium Chloride (TTC) 

On day 4 after surgery, the mice were euthanized. The brain was quickly removed, and using a Brain Matrix Slicer (Zivic instruments, Pittsburgh, PA, USA), it was sectioned into 2 mm thick slices starting at the frontal pole. The slices were submerged in 2% TTC (J.C. Baker, Coimbatore, India) in a Petri dish and incubated at 37 °C for 5 min. Mitochondrial dehydrogenases convert TTC, a water-soluble salt, into formazan, which is a red, lipid-soluble compound that stains normal tissue deep red [31,32]. Ischemic tissue in which the mitochondrial function was abnormal was identified using reduced TTC stain [33]. TTC-stained slices (2, 4, 6, 8, and 10 mm from the frontal pole) were scanned using a scanner to evaluate lesion volume and then analyzed using Image-J analysis software 1.52j [34] (public domain software developed at NIH and available on the Internet at http://imagej.nih.gov/ij/dowload.html (accessed on 29 December 2018)). The percent of the total combined hemispheric volume was used to determine lesion volume and was calculated as:[(Vc − VL)/Vc] · 100
where Vc is the volume of both hemispheres (compare the value with the whole brain of sham) and VL is the volume of non-lesioned tissue in the lesioned hemisphere [34,35]. Then, these sections were analyzed in the different treated and untreated animal groups.

#### 2.3.10. Fate of the Surviving Animals at the Conclusion of the Experiment

There were no additional animals remaining at the end of the experiment.

### 2.4. Sample Collection for the Western Blot Analysis

RIPA lysis and extraction buffer (25 mM Tris_HCl pH 7.6, 150 mM NaCl, 1% NP-40, 1% sodium deoxy-cholate, 0.1% SDS) (G-Biosciences, St. Louis, MO 63132-1429, USA cat # 786-490) containing 1% (*v*/*v*) mammalian protease inhibitor cocktail and 1% (*v*/*v*) phosphatase inhibitor cocktail, from Sigma and Thermo Scientific, respectively, were used to lyse PC-12 cells [5,11]. A Bradford protein assay (Bio-red) was used to analyze the protein concentrations in each sample, and the samples were stored at −80 °C until use.

Isoflurane (Phoenix) was used to anesthetize animals, and their brains were rapidly removed and kept on ice after decapitation. For immunoblotting, the left hemisphere [36] and the right hemisphere (identical corresponding parts) were quickly cut apart on dry ice and homogenized in lysis buffer consisting of 50 mM Tris–HCl, 150 mM NaCl, 2 mM EDTA, pH 8.0, 1% Triton-X-100, 1:100 dilution of mammalian protease inhibitors (Sigma-Aldrich, St. Louis, MO, USA), and protease inhibitor [37]. A Bradford protein assay was used to calculate protein concentration in each sample, and the samples were stored at −80 °C until use.

A 12% sodium dodecyl sulfate–polyacrylamide gel electrophoresis was used to separate protein samples, which were transferred onto nitrocellulose membranes. A Western blot was conducted as described before [37] with the following primary antibodies overnight: Abcam (Waltham, MA 02453, USA): GRP 78, Hsp 70, Hsp 27, activating transcription factor 4 (ATF4), Caspase- 12, IRE1 and X-box-binding protein 1 (XBP-1) (1:500); Cell Signaling(Danvers, MA 01923, USA): GAPDH (1:3000), Beclin 1, eukaryotic translation initiation factor 2 α (eIF2α), P-eIF2α, P-STAT 3, Bax, AKT, phosphorylated AKT (p-AKT) and Cleaved Caspase-3 (1:500); Santa Cruz: CHOP/GADD153) and Bcl2 (1:500); Novus Biologicals (Centennial, CO 80112, USA): ATF6 (1:1500) (Appendix A). The membranes were washed three times with Tris-buffered saline containing 0.1% Tween-20 (TBS-T) and incubated with secondary antibodies for 1 h at room temperature. The secondary antibodies utilized were goat IRDye 800-conjugated anti-rabbit (1:15,000) and IRDye 680 conjugated anti-mouse (1: 15,000) antibodies (LI-COR Biosciences, Lincoln, NE, USA). Fluorescent signals were identified with an LI-COR Odyssey Fc system, and the images were analyzed with either Image Studio 2.0 software or Image J software 1.52j [36].

### 2.5. Dosage of Carbamathione

We used different dosages of Carb for various assessments. We used a 25 µM dose for in vitro studies, a 200 µM dose for whole-cell patch clamping, and for in vivo studies, 14 mg/kg was used in all experiments.

### 2.6. Data and Statistical Analysis

The mean ± SEM was used to express all data. For statistical analysis, the SPSS computer program (SPSS 15.0, Chicago, IL, USA, and Prism Graph Pad 8) was used. A minimum of N = 5 replicates per condition were performed for each in vitro cell culture experiment and 5 independent experiments were conducted for the purpose of statistical significance using ANOVA. A *t*-test or ANOVA combined with the Dunnett post hoc test was used to determine the statistical significance between groups. An assessment of the normality of the data was conducted using an SPSS Shapiro–Wilk test. Differences of *p* < 0.05 were considered statistically significant. For each experiment, at least three independent replicates were performed and analyzed after the omission of outliers.

## 3. Results

### 3.1. Glutamate Excitotoxicity Is Dose-Dependent in PC-12 Cell Culture

Cell viability of PC12 cells decreased under conditions of 24 h exposure to glutamate, in a range from 0.01 mM to 150 mM (Figure 1a). We chose the optimal dose of 10 mM glutamate for the cell viability test [23,25]. At 10 mM glutamate, approximately 45% survival of PC-12 cells was observed with a range from 76% at 10 µM glutamate to 18% at 150 mM glutamate (*F* (9, 30) = 555.6; *p* < 0.01, *(n =* 6); *ANOVA with Dunnett post hoc test*).

### 3.2. Carbamathione Protects PC-12 Cells against Glutamate-Induced Excitotoxicity

In our previous research, it was determined that pre-incubation with 25 µM DETC MeSO resulted in maximal recovery from neuronal injury induced by glutamate [6]. To determine the protective effect, we used 0.5 µM to 500 µM concentrations of Carb with 10 mM glutamate-induced excitotoxicity, and cell viability was examined using an ATP assay, as shown in Figure 1b. The result showed that 10 mM glutamate significantly decreased the survival of PC-12 cells to about 50–55%. The protective effect of Carb was up to a level of 1.3-fold and 1.6-fold compared to the no-drug condition, as shown in Figure 1b, after treatment with 25 µM Carb for 1 h (*F* (10, 47) = 93.97; *p* < 0.01 (*n* = 6)) and 24 h (*F* (10, 44) = 28.96; *p* < 0.01 (*n* = 6); *ANOVA with Dunnett post hoc test for both 1 h and 24 h*) respectively, following exposure to 10 mM glutamate for 24 h.

#### 3.2.1. Carbamathione Modulates Expression of Heat Shock Protein (Hsp) and AKT Induced by Glutamate Toxicity

In many cell types, Hsp 27 has been shown to control apoptosis by modulating AKT activation. A previous study identified Hsp 27 as an AKT substrate that detaches from AKT upon AKT phosphorylation [38]. Carbamathione resulted in increased expression of Hsp 27 by 3-fold in 1 h and by 3.5-fold in 24 h in the Carbamathione-treated groups relative to the glutamate (10 mM)-treated groups (Figure 2a; *F* (3, 22) = 10.47; *p* < 0.05 (*n* = 6)). AKT, also known as protein kinase B (PKB; a serine/threonine-specific protein kinase) plays a critical part in modulating survival and apoptosis. In our experiments, P-AKT (the activated form of AKT) showed a dramatic up-regulation of 2.5-fold in 1 h and 2.75-fold in 24 h in the Carbamathione-treated groups, respectively, related to the glutamate (10 mM)-treated groups (Figure 2b; *F* (3, 13) = 16.65; *p* < 0.01 (*n* = 5)), whereas AKT expression showed no significant changes when Carbamathione was used (Figure 2b; *F* (3, 20) = 0.558, *p* = 0.6488 (*n* = 5)). Importantly, Carbamathione treatment increased the ratio of P-AKT to AKT by 2.2-fold in the 1 h and by 2.6-fold in the 24 h groups relative to the glutamate (10 mM)-treated groups (Figure 2c), confirming a highly protective role for the drug.

#### 3.2.2. Effect of Carbamathione on the Expression Beclin 1, a Marker for Autophagy

The autophagy marker [14], Beclin 1, shows a very clear decreasing trend in the Carbamathione treatment for both the 1 h and 24 h drug-treated groups to <50% relative to the glutamate-treated group, as shown in Figure 3a,b (*F* (3, 17) = 9.209; *p <* 0.01 (*n* = 6)). Our data indicate that Carb reduces the need for the cell to digest/destroy its components (as cellular self-digestion occurs when a cell is under stress).

#### 3.2.3. Carbamathione Can Decrease Apoptosis by the Down-Regulation of Apoptotic Markers

Cell survival pathways are initiated by the Bcl2 protein, which in turn inhibits mitochondrial permeability transition pore (MPTP) opening, while Bax, BAD, and/or Bak, which are all proapoptotic proteins, can translocate from the cytosol into the outer mitochondrial membrane after glutamate toxicity to cause MPTP opening and cytochrome C (Cyt-c) release. Cyt-c triggers caspase-3, which is assumed to be the final step of apoptosis [39]. Our results show that Carb down-regulated proapoptotic protein Bax in the drug-treated groups (to <30% in 1 h and to <40% in 24 h) relative to the glutamate (10 mM)-treated groups (Figure 3d; *F* (3, 12) = 10.21; *p <* 0.01 (*n* = 5)). On the other hand, measurement of anti-apoptotic protein Bcl2 showed an increase in the ratio of Bcl2/Bax with Carb (by 2-fold in 1 h and 2-fold in 24 h) relative to the glutamate (10 mM)-treated group (Figure 3c (*F* (3, 18) *=* 12.87; *p* < 0.01 (*n* = 5) and Figure 3e (*F* (3, 18) = 29.01; *p* < 0.01 (*n* = 5)). Up-regulation of Bcl2 could decrease cleaved caspase-3 in the Carb-treated (1 h and 24 h) group to <30% and <40%, respectively, in comparison to the glutamate (10 mM)-treated group (Figure 3f (*F* (3, 9) = 6.722; *p <* 0.05 (*n* = 5); *ANOVA with Dunnett post hoc test for*
Figure 3a–f).

#### 3.2.4. Carbamathione Protects Neuronal Cells against Glutamate Excitotoxicity by Suppressing the Expression of GRP 78 and CHOP

To examine if ER stress can be caused by glutamate and then inhibited by Carb, specific ER stress effector proteins were analyzed using a Western blot. Glucose-regulated protein-78 (GRP 78) is an ER-associated chaperone, which enhances protein folding in the ER [5]. Experimental evidence indicated that up-regulation of GRP 78 stops neuronal damage induced by ER stress, and the increase in GRP 78 expression may be related to the degree of neuroprotection [34]. The expression of GRP 78 protein was down-regulated to <70% relative to the control group in primary neurons after treatment with 10 mM glutamate for 24 h. However, Carb increased the level of GRP 78 back to control levels by 1.7-fold in 1 h and 1.85-fold in 24 h relative to the glutamate-treated group, as shown in Figure 4a (*F* (3, 22) = 10.24; *p <* 0.01 (*n* = 5)). C/EBP homologous protein (CHOP), also referred to as growth arrest and DNA damage-inducible protein 153 (GADD153), is a chief ER stress marker [40]. Figure 4b shows that the expression of CHOP was upregulated 1.5-fold in the glutamate group compared to the control group. Carb treatment restored CHOP expression to <60% in 1 h and <70% in 24 h relative to the glutamate-treated group (Figure 4b (*F* (3, 13) = 3.530; *p* < 0.05 (*n* = 5)).

### 3.3. Carbamathione Demonstrates Robust Protective Activity against Hypoxia/Reoxygenation in PC-12 Cell Cultures

This in vitro study was designed to determine the efficacy of Carb in exerting protection against cell stress using an in vitro hypoxia/reoxygenation model of stroke. Hypoxia/reoxygenation leads to a reduction in the calcium levels in the ER, resulting in the disruption of ER homeostasis and the resulting accumulation of unfolded/missed folded proteins. To determine the appropriate concentration of Carb in cultures, PC-12 cells were exposed to hypoxia and reoxygenation in the presence or absence of 5–100 µM Carb, as shown in Figure 1c (*F* (11, 48) = 7.956; *p* < 0.01 (*n* = 6); *ANOVA with Dunnett post hoc test*). After hypoxia and reoxygenation, ATP levels for cells without Carb treatment dropped to about 47% (percentage of control). Carb treatment dramatically increased cell viability. The presence of 25 µM Carb clearly improved cell viability to greater than 70%. The maximal protective effect of Carb was at 10 and 25 micromolar in hypoxia/reoxygenation and at 25 micromolar with glutamate. H/R doses above 100 micromolar may be toxic due to excessive levels of Carb. Our data showed that 25 μM Carb can attenuate cell death in hypoxia/reoxygenation. Our comparison between the effects of Carb on glutamate and hypoxia/reoxygenation conditions showed similar clear protection in the range of 10 to 50 micromolar of Carb.

#### 3.3.1. Effect of Carbamathione on the Expression of Heat Shock Protein, AKT, and P-STAT 3 Induced by Hypoxia/Reoxygenation

Mammalian cells can react to a range of stresses such as heat, cold, oxidative stress, metabolic disturbance, and environmental toxins through necrotic or apoptotic cell death, while increased expression and phosphorylation of heat shock proteins such as Hsp 27 can guard cells against cellular stress. Heat shock proteins (Hsp) generally display molecular chaperone activity and act together with a broad range of proteins to exhibit specific effects. Previously, Settler et al. [41] showed that Hsp 27 overexpression confers long-term neuroprotection against cerebral ischemia. Increases in the levels of Hsp 70 by 4.25-fold and Hsp 27 by 3-fold were found in the Carb-treated group relative to the hypoxia–reoxygenation group alone (Figure 5a (*F* (2, 14) = 6.364; *p* = 0.05 (*n* = 5)) and 5b (*F* (2, 8) = 189.6; *p* < 0.01 (*n* = 5)). In this experiment, P-AKT (the activated form of AKT) showed up-regulation by 1.5-fold in the Carb-treated groups relative to the hypoxia–reoxygenation-treated groups (Figure 5c (*F* (2, 9) = 4.974; *p* < 0.05 (*n* = 5)), whereas AKT expression showed no significant changes when Carb was used (Figure 5c (*F* (2, 10) = 0.3336; *p* = 0.7240 (*n* = 5)). The expression of P-STAT 3 was determined using a Western blot (Figure 5d (*F* (2, 12) = 4.867; *p* < 0.05 (*n* = 5)). Carb significantly decreased P-STAT 3 to <70% in the Carb-treated groups relative to the hypoxia/reoxygenation-treated groups.

#### 3.3.2. Carbamathione Inhibits the Expression of GRP 78, CHOP, and Caspase-12 Induced by Hypoxia/Reoxygenation

To observe the effect of Carb on the ER stress induced by hypoxia/reoxygenation, we preincubated 25 µM Carb for 1 h followed by hypoxia and reoxygenation. The expression levels of GRP 78 and CHOP were measured using a Western blot analysis, as shown in Figure 4c,d. The expression of GRP 78 increased by 1.4-fold and CHOP increased by 1.25-fold after exposure to hypoxia/reoxygenation (Figure 4c (*F* (2, 19) = 4.686; *p* < 0.05 (*n* = 7)) and Figure 4d (*F* (2, 7) = 5.904; *p* < 0.05 (*n* = 5)). The Western blot analysis showed that the levels of caspase-12 (42 kDa) increased by 2.5-fold and cleaved caspase-12 (38 kDa) increased by 1.3-fold after hypoxia/reoxygenation (Figure 6j *(F* (2, 13) = 27.82; *p* < 0.01 (*n* = 6)) and Figure 6k (*F* (2, 21) = 5.090; *p* = 0.01 (*n* = 7)). Carb significantly reduced the expression of GRP 78 to <70%, CHOP to <50%, caspase-12 to <40%, and cleaved caspase-12 to <40% in the Carb-treated groups relative to the hypoxia–reoxygenation-treated groups, demonstrating that Carb has the ability to inhibit the apoptosis induced by ER stress in hypoxia/reoxygenation.

#### 3.3.3. PERK and IRE1 Pathways Were Inhibited by Carbamathione under Hypoxia/Reoxygenation, although the ATF-6 Pathway Was Activated

It is fully recognized that there are three ER stress-induced signaling pathways: the PERK, ATF6, and IRE1 pathways. Since Carb can protect cells against ER stress induced by hypoxia, we aimed to further classify which signaling pathway is implicated in the protective process. P-eIF2α and ATF4 were highly expressed after hypoxia/reoxygenation, where ATF4 increased by approximately 2.0-fold over control. However, after treatment with Carb followed by hypoxia/reoxygenation, the levels decreased in the cells: ATF 4 to <30% and eIF2α to <60% (Figure 6d (*F* (2, 6) = 131.1; *p* < 0.01 (*n* = 5)) and Figure 6e ((*F* (2, 7) = 25.90; *p* < 0.01 (*n* = 5)), indicating that Carb inhibits the activation of the PERK pathway under this condition. This result indicated that Carb has observable effects on PERK pathway activation. We next examined the effect of Carb on the ATF6 pathway in cells induced by hypoxia/reoxygenation. Treatment with Carb reduced the level of ATF6 (90 kDa) to <55% (Figure 6g ((*F* (2, 18) = 9.626; *p* < 0.01 (*n* = 5)). However, the cleaved ATF6 (50 kDa) in cells treated with Carb increased 1.8-fold relative to cells under hypoxia/reoxygenation without Carb, as shown in Figure 6h (*F* (2, 9) = 40.42; *p* < 0.01 (*n* = 5)) and Figure 6i (*F* (2, 21) *=* 27.62; *p* < 0.01 (*n* = 5)). These results demonstrate that Carb cannot prevent the activation of the ATF6 pathway under hypoxia/reoxygenation. To determine if Carb affects the IRE1 pathway induced by hypoxia/reoxygenation, we tested the expression of IRE1 and XBP1 in PC-12 cells with and without Carb treatment under hypoxia/reoxygenation conditions using a Western blot analysis (Figure 6b ((*F* (2, 10) = 23.99; *p* < 0.01 (*n* = 5)) and Figure 6c (*F* (2, 13) = 30.35; *p* < 0.01 (*n* = 5)). The results showed elevated expression of IRE1 by 2.5-fold and XBP1 by 2-fold in cells under hypoxia/reoxygenation. Carb reversed the increased expression of IRE1 to <70% and XBP1 to <30% relative to normal control levels, demonstrating that Carb significantly inhibits the IRE1 pathway in ER stress induced by hypoxia/reoxygenation. Considering information on all three ER stress pathways our, data indicate that Carb inhibits the IRE-1 and PERK pathways but activates the ATF-6 pathway. In light of the fact that Carb inhibits Grp78, CHOP, and caspase 12 as well as the decreased activity of the IRE-1 and PERK pathways, our data point overall to a decreased ER stress response that results functionally in a pro-survival effect elicited by Carb.

#### 3.3.4. Effect of Carbamathione on the Hypoxia/Reoxygenation-Induced Change in Beclin 1, Bcl2, Bax, and Caspase-3 Expression

A number of studies have reported that a cellular biomarker for autophagy, Beclin 1, is upregulated in ischemic brain injuries [42]. Our data showed that Carb had a positive effect on reducing the level of Beclin 1 to <75% in the Carb-treated groups relative to the hypoxia/reoxygenation-treated groups (Figure 7b (*F* (2, 7) = 12.54; *p* < 0.01 (*n* = 5)). The Bcl2 family members, Bcl2 and Bax, are key regulators of apoptosis. Pro-apoptotic proteins such as Bax are initially limited to the cytosol [43]. The death signals direct pro-apoptotic proteins to translocate into the mitochondrial membrane to form homodimers of Bax and hence target mitochondria initiating the release of cytochrome c [44]. Antiapoptotic proteins including Bcl2 are primarily integral membrane proteins [45] and can form heterodimers with pro-apoptotic proteins [46] and avoid the increase in homodimers of Bax, resulting in inhibiting the release of cytochrome c from mitochondria. Thus, a rise in the ratio of Bcl2: Bax would result in protective effects. Cells were treated with hypoxia/reoxygenation or with hypoxia/reoxygenation in the presence of 25 µM Carb and then harvested to determine the levels of Bcl2 and Bax protein expression using a Western blot analysis. As shown in Figure 7c, top lane 2, and Figure 7d, lane 2, hypoxia/reoxygenation caused a down-regulation of Bcl2 to <80% relative to the normal control. When the cells were preincubated with Carb, this reduction in Bcl2 was returned to 115% of the normal control. (Figure 7a,c (*F* (2, 7) = 4.698; *p* < 0.05 (*n* = 5)). In contrast, hypoxia/reoxygenation-treated cells show an up-regulation of Bax by 1.75-fold when compared with the control groups, and this up-regulation was reversed in cells preincubated with Carb to 110% of the sham no-drug levels (Figure 7d (*F* (2, 12) = 7.555; *p* < 0.01 (*n* = 5)). As shown in Figure 7e, lane 2, the hypoxia/reoxygenation-induced changes in Bcl2 and Bax levels resulted in a decrease in the Bcl2: Bax ratio to <50%. In contrast, in cells preincubated with Carb, the Bcl2: Bax ratio was approximately 2.25-fold higher, confirming the protective nature of Carb exposure (Figure 7e (*F* (2, 12) = 12.43; *p* = 0.01 (*n* = 5)), lane 3). Up-regulated Bcl2 decreased cytochrome c release, which decreased cleaved caspase-3 to <40% in the Carb-treated group compared to the control group (Figure 7f (*F* (2, 14) = 3.711; *p* = 0.01 (*n* = 5)), lane 3 compared to lane 2).

### 3.4. The Effects of Carb on the Suppression of NMDA Glutamate Receptor Activation in Retinal Neurons

To investigate the direct effects of Carb on the suppression of NMDA glutamate receptor activation, we performed whole-cell patch-clamp recordings of glutamate receptor currents in mouse retinal ganglion cells using a whole-mount preparation. The synaptic glutamate currents in the ganglion cells were evoked with a light stimulus to activate the glutamate transmission pathway from photoreceptors to ganglion cells. Figure 8 shows the results for light-evoked glutamatergic currents in a ganglion cell (see black trace). It is known that glutamate activates both NMDA and non-NMDA receptors (also called kainite and AMPA receptors) that are the major synaptic receptors in retinas, and cyanquixaline (CNQX) is a specific antagonist for non-NMDA glutamate receptors. We applied 100 μM CNQX to block non-NMDA receptors in the light-evoked glutamate currents (see blue trace); the remaining CNQX-insensitive current was an NMDA glutamate receptor current that could be inhibited by 200 μM Carb (see red trace), indicating that Carb can reduce the activation of NMDA receptors (Figure 8). These results suggest that Carb might have neuroprotective effects against NMDA glutamate receptor-mediated secondary neural degeneration after a stroke.

### 3.5. Carbamathione Treatment Attenuates Infarction Volume and Neurological Deficits

While the mouse model closely replicates human global cerebral ischemia, there have been reports of minor differences in cerebrovascular morphology between rodents and humans, which may include the internal carotid artery, the circle of Willis, and the posterior cerebral artery. There are also differences in the number of collaterals between anterior and middle cerebral arteries, which are higher in rodents than in humans [47,48]. 

The findings of this study can generalize to other species regarding the effects of global cerebral ischemia and would generalize to effects in the penumbra of focal cerebral ischemia models. This in vivo study was designed to determine the efficacy of Carb in exerting protection against global ischemia in a mouse model. The following hypotheses were tested: Carb would exert protection against ER stress-induced apoptosis in the global ischemic brain by down-regulating CHOP, GRP 78, and Bax.

In our in vivo study, in addition to evaluating the overall advancement in reducing the brain infarct size, we also analyzed the infarct size at 2, 4, 6, 8, and 10 mm from the frontal pole to evaluate the effect of ischemia and Carb in different brain sections, as shown in Figure 9a–c. The TTC staining for mice subjected to BCAO in the vehicle-treated group versus the Carb-treated groups is shown in Figure 9a–c. The mean infarct volumes were noticeably reduced to <50% in the mice treated with Carb compared to the vehicle-treated groups (Figure 9c (*t* = 2.716, *df* = 8; *p* < 0.05, (*n* = 9)). No infarction was observed in the sham group. Carb markedly reduced the volume of the lesion in sections 2, 4, 6, 8, and 10 mm when Carb was injected 30 min after occlusion. The sham-operated group showed no ischemic injury, as determined using TTC staining.

Unlike mice with brain injury, when normal animals run into a corner, they naturally rear forward and upward and then turn back, in either direction, to face away from the corner and toward the open end of the setup (the corner test) [27]. Hence, an animal’s asymmetric direction of turning when encountering a corner is used as an indicator of brain injury. We used an experimental corner setup composed of two boards (30 × 20 × 1 cm^3^) arranged to form a 30° corner. We measured the rate of turns when normal mice faced a 30° corner (i.e., turning to either the left or right) to be 50 ± 8% (symmetric and no bias) before the BCAO-30 min surgery. Mice were tested before sacrifice 4 days after the BCAO-30 procedure. We observed a significant increase in asymmetric turning (~90% to one side) in BCAO-30 mice when facing a 30° corner. BCAO mice without treatment showed significant asymmetric turning (~90%) beginning one day after the procedure and persisting for at least four days (Figure 9g,h *((F* (2, 45) = 4.114; *p* < 0.05 (*n* = 9)). Mice with treatment (BCAO + Carb) exhibited behavior that was not significantly different from that seen in the sham-operated mice.

To test whether motor dysfunction might contribute to the behavioral deficits observed in the corner test, the locomotor activity of mice was measured using a force-plate actometer [28,29]. We observed similar differences in behavior between the sham group and the BCAO with Carb-treated group. The travel distances, the number of the sham group and the BCAO with Carb group were significantly different compared to the BCAO with vehicle group (Figure 9e,f *(F* (2, 33) = 17.90, (*n* = 9); *p* < 0.01), distance: sham: 151.25 ± 8.39 m; BCAO with vehicle: 100 ± 3.90 m; BCAO with Carb 148.15 ± 7.45, *n* = 9, *p <* 0.05).

#### Carbamathione Can Modulate the Unfolded Protein Response and Decrease Apoptosis by Down-Regulating Apoptotic Markers

Our data showed that after BCAO, in the Carb-treated group, GRP 78 dramatically decreased after 4 days compared to the vehicle-treated group. GRP 78 is commonly used as an indicator for the UPR. As we can see in Figure 10a *(F* (5, 58) = 7.959; *p* < 0.01, *n* = 6), Carb decreased the expression of GRP 78 to less than 65% relative to the vehicle-treated groups. ATF4, XBP1, and ATF6 all functionally converge on the promoter of the gene encoding the protein CHOP, which stimulates transcription of the mRNAs encoding Bim and BCL2 [6]. p38MAPK plays an important role in stimulating CHOP activity. JNK activates BIM but inhibits BCL2. We measured the expression of CHOP with a Western blot analysis using the BCAO stroke model. As shown in Figure 9b, the expression of CHOP was up-regulated 3-fold in the BCAO model in comparison to the sham-operated group. The Western blot analyses showed that Carb significantly decreased the levels of CHOP to <50% in the Carb-treated group compared to the vehicle-treated BCAO group (Figure 10b *(F* (5, 42) = 10.13; *p* < 0.01, *n* = 6).

Given that Bcl2 and Bax are downstream targets of cell death pathways, consistent with the previous studies, ischemia induced a significant increase in the proapoptotic protein Bax and a decrease in the antiapoptotic protein Bcl2. Compared with the vehicle treatment, Carb significantly increased the expression of Bcl2 2-fold and decreased Bax to <50% of the protein level on day 4 (Figure 10c (*F* (5, 55) = 3.627; *p* = 0.01, *n* = 6) and Figure 10d *(F* (5, 52) = 8.133; *p* < 0.01, *n* = 6)). Our results indicate that measuring the ratio of pro-survival Bcl2 to pro-death Bax on day 4 displayed a major up-regulation of Bcl2/Bax (greater than 3-fold up-regulation) following Carb treatment (Figure 10e ((*F* (5, 56) = 3.402; *p* < 0.01, *n* = 6)).

## 4. Discussion

Despite extensive research into developing medicines for stroke based on the established methods, either as glutamate receptor antagonists, Ca^2+^ channel blockers, enzyme inhibitors, inhibitors of apoptotic pathways, or ROS scavengers, these attempts have been unsatisfactory [49]. Complexity in the molecular ischemic cascade is perhaps one of the important reasons for the failure of many clinical trials [50]. Therapeutic interventions targeted at blocking individual molecular pathways have proved to be inadequate for the task. It is generally believed that one of the leading causes of brain injury in stroke is glutamate-induced excitotoxicity resulting in a marked increase in intracellular calcium. Glutamate neurotransmission is essential for normal neuronal function; hence, glutamate receptor antagonists that completely block glutamate receptors would have a detrimental effect. Here, we reported the development of a potential therapeutic agent for stroke treatment targeted at the sites that modulate, but not totally block, glutamate neurotransmission. The therapeutic agent that we report here is a glutamate receptor partial antagonist, S-(N, N-diethylcarbamoyl)glutathione, known as Carbamathione (Carb). The rationale behind using Carb for stroke intervention is based on the following observation: In this report, showed that Carb, an NMDA receptor partial antagonist [17,22] and an active metabolite of disulfiram (DSF), has a potent neuroprotective function against hypoxia-induced or ischemia-induced brain injury in cell-based or animal-based stroke models. Using a cell-based stroke model, we found that Carb is effective in protecting PC12 cells against glutamate-induced (Figure 1b) or hypoxia-induced cell injury (Figure 1c), which is generally accepted as the mechanism underlying ischemic stroke-induced brain injury. The mechanism underlying the protection of Carb is in part due to its partial antagonistic action on glutamate receptors, especially the NMDA receptors, resulting in the activation of the AKT signaling pathway, leading to increased expression of pro-cell survival biomarkers such as Hsp 27 (Figure 2a), P-AKT (Figure 2b,c), and Bcl2 (Figure 3c) and decreased expression of pro-cell death markers such as Beclin 1 (Figure 3b), Bax (Figure 3d), cleaved caspase-3 (Figure 3f), and ER stress markers, e.g., CHOP (Figure 4b,d), IRE1 (Figure 6b), XBP1 (Figure 6c), ATF 4 (Figure 6d), and eIF2α (Figure 6e).

The in vitro experiments provide relatively quick data that are both important and crucial for understanding a specific area of research. However, they tend to be less physiologically applicable than experiments carried out in vivo because they do not demonstrate the real biological conditions of a living organism [51]. Moreover, the reaction of the brain to stroke and the effects resulting from other systems make it impossible to mimic stroke using only in vitro systems [52]. The studies on stroke using animal models allow accurate control over the severity, duration, location, and cause of the ischemia and facilitate control of physiological parameters, such as body temperature [53], all of which create a more similar scenario to human conditions. PC12 cells may not totally replicate the phenotype of primary cultured neurons, and for this reason, our work would also need to use in vivo models. Hence, the data obtained from in vivo studies are more easily inferred to human conditions than those obtained from in vitro studies.

In the stroke BCAO animal model, we found that Carb greatly reduced the size of the brain infarction (Figure 9c) and markedly improved the functional outcome at the molecular/cellular level, including a decrease in ER stress markers GRP 78 (Figure 9a) and CHOP (Figure 9b), and at behavioral levels, such as a decrease in asymmetric turns (Figure 9g,h) and an increase in locomotor activity (Figure 9e,f). These observations reported in this paper are consistent with our previous report, which found that DETC-MeSO, a precursor of Carb, has a neuroprotective function in stroke using animal-based as well as cell-based models [54,55].

Previous studies demonstrate that Carb blocks glutamate synapses in the brain, suggesting a key role for Carb in inhibiting the action of glutamate in the CNS [17,21]. In the current study, we specifically examined the mechanisms of action of Carb on neuroprotective function and tested the role of Carb in inhibiting the activation of NMDA receptors. Our data on mouse retinal ganglion cells clearly demonstrate that Carb is a partial antagonist for NMDA glutamate receptors. These data provide important information suggesting that the effect of Carb administration in reducing neurotoxicity in the stroke model can occur via inhibition of NMDA receptors.

Carb and DETC-MeSO both have antagonistic effects on brain glutamate receptors [22]. Disulfiram exerts its anti-alcohol effect only after bio-activation to the active metabolite DETC-MeSO [56]. As shown earlier, DETC-MeSO is a potent and selective carbamoylating agent for sulfhydryl groups in glutamate receptors [57]. The carbamoylation of glutathione by DETC-MeSO leads to the formation of S-(N, N-diethylcarbamoyl) glutathione (Carbamathione). The percentage of Carb formed from disulfiram is unknown but presumed to be small, at less than 0.05% [58]. The parent drug DETC-MeSO partially blocks glutamate binding to a synaptic membrane preparation from the brain and prevents seizures in mice induced by glutamate analogs and hyperbaric oxygen [17]. A high increase in glutamate concentration was demonstrated in several neurodegenerative diseases including Alzheimer’s disease [59] and Huntington’s disease [60] as well as in stroke [61]. One of the pathways involved in glutamate-induced cell death is calcium overload [62]. Two kinds of glutamate receptors, i.e., the ionotropic and metabotropic receptors, produced glutamate’s excitatory effects [63]. NMDA glutamate receptors are believed to be the crucial mediators of death during excitotoxic injury [6,63]. During ischemia and throughout glutamate excitotoxicity, Ca^2+^ influx through NMDA receptors promotes cell death more efficiently than through other types of Ca^2+^ channels [63].

It is known that brain ischemia followed by glutamate excitotoxicity leads to intracellular calcium overload and initiates a series of intracellular events, such as the release of apoptotic proteins leading to apoptotic cell death [64]. As we hypothesized that Carb can regulate intracellular calcium homeostasis by preventing Ca^2+^ influx, we continued testing mitochondrial function by measuring different pro- and anti-apoptotic markers. Our results demonstrated that Carb can down-regulate the proapoptotic protein Bax. On the other hand, measurements for the anti-apoptotic protein BCL2 showed an increased ratio of BCL2/Bax in the Carb-treated PC-12 cells and in the brain of Carb-treated versus vehicle-treated groups. A high ratio of Bcl2 to Bax can prevent the release of cytochrome-c from mitochondria, which results in decreased caspase-3 activity and helps in survival. Carb can decrease caspase-3 activation in the glutamate-treated and hypoxia/reoxygenation-treated PC-12 cell. The essential autophagic initiator Beclin 1 is upregulated during reperfusion and causes autophagic cell death [14,65].

The harmful autophagic activation facilitated by Beclin 1 is correlated with Bcl2 downregulation as Bcl2 inhibits Beclin 1 function by binding to only the BH3 domain of Beclin 1 [66]. We observed the upregulation of Beclin 1 after glutamate and hypoxia/reoxygenation injury. However, Beclin 1’s harmful influence was counteracted by the effect of Carb to upregulate Bcl2, thereby rescuing the cell from autophagic death.

Up-regulation of Hsp 27 and Hsp 70 mRNA and protein after a variety of insults has been demonstrated in neuronal cultures and brain tissue extracted from animals exposed to stress [67]. In vitro evidence from HSP-transfected neurons confirms the protective role of Hsp 27 and Hsp 70. Trials using in vivo models of epilepsy and stroke indicate that transgenic overexpression or virally generated HSPs are not always able to lower lesion size but can augment cell survival [68]. Since Hsp 27 and Hsp 70 are the main inducible HSPs in the central nervous system [69], we investigated the effect of Carb using in vitro models of stroke (specifically, glutamate excitotoxicity and hypoxia/reoxygenation) on these two HSPs using Western blotting. Hsp 27 and Hsp 70 showed decreased expression in cells of vehicle-treated glutamate or hypoxic/reoxygenation groups in comparison to the control group. Applying Carb increased the expression of Hsp 27 in both glutamate and hypoxia/reoxygenation conditions, relative to “no drug” controls. Our data also indicated that HSP70 changed in a similar manner to Hsp27. Our finding that Carb changed Hsp 70 expression adds strength to the concept that elevated Hsp70 may be signaling for specialized protective activities through its chaperone functions. In this study, increasing Hsp 27 expression might be more protective due to its links with cytoskeletal stability and its ability to repair proteins in an ATP-independent fashion in conjunction with its anti-apoptotic functions. Hsp 27 was reported to protect cells in vitro by affecting both caspase-dependent and caspase-independent apoptotic pathways. Some reports also show that caspase-3 activation after ischemia is blocked by heat shock proteins [69]. Surprisingly, substituting Hsp 27 in rat neurons for a caspase-blocking agent does not stop cytotoxicity, indicating that, in addition to caspase-3 blockage, Hsp 27 must use chaperone functions that stop necrosis and caspase-independent cell death [70]. A key kinase known to inhibit apoptosis is protein K = kinase B (AKT). Many studies have indicated that activated AKT (P-AKT) augments neuroprotection during cerebral ischemia [71]. We observed that Carb activates AKT, thereby eliciting protection against glutamate excitotoxicity and hypoxia/reoxygenation injury. The use of Carb caused a markedly increased level of P-AKT expression in the cells of the Carb-treated group. The cellular decision to undergo apoptosis is regulated by the integration of multiple survival and death signals. The AKT serine/threonine kinases are major mediators of cell survival in response to Ca^2+^ influx [71]. P-AKT also has been shown to activate some anti-apoptotic markers, such as BCL2 and mTOR [72]. Our data on the pro-apoptotic marker Bax and the anti-apoptotic marker BCL2 support the role of P-AKT up-regulation in protecting cells in the Carb-treated group.

In the CNS, the STAT3 signaling pathway has been reported to play a vital role in inflammatory responses [73]. STAT3 dimerization, nuclear translocation, and DNA binding elicit the transcription of genes encoding a number of inflammatory factors when STAT3 is phosphorylated at Tyr705 [74,75,76]. In our study, we observed that P-STAT3 was up-regulated after hypoxia/reoxygenation injury, and this effect was reversed with Carb treatment.

The ER is the location where many pathological processes such as alterations in calcium homeostasis, glucose deprivation, and hypoxia occur. Primarily in response to stressful stimuli, the ER will evoke the UPR, which is a pro-survival signal transduction pathway [77]. GRP 78 is known as a key ER chaperone, and it is in the ER lumen and is also used as a transmembrane protein. In addition, it is found outside the ER and acts as an important marker for ER stress [78]. As we reported in the Results section, we detected changes in GRP 78 expression after Carb treatment that protected against hypoxia/reoxygenation injury. However, in glutamate excitotoxicity conditions, Carb increased the expression of GRP 78 in comparison to the glutamate-treated control group. Based on our previous studies on Sulindac and DETC MeSO [6,34], it was shown that ER stress is reduced by ischemic preconditioning pathways and that ER stress reduction by preconditioning was the result of ER molecular chaperone induction [79]. Depending on the levels and timing of expression, GRP 78, therefore, serves as a molecular chaperone in glutamate excitotoxicity and may also serve as an ER stress marker in hypoxia/reoxygenation injury and in BCAO, specifically in our experiments where Carb treatment was used. Following ER stress and accumulation of unfolded proteins, GRP 78 dissociates from each sensor (ATF6, PERK, and IRE-1 inside the ER lumen) and proceeds to cause the UPR. In this paper, our primary aim was to identify which ER stress-induced pathway could be affected with Carb treatment during the process of hypoxia/reoxygenation exposure in the PC-12 cell culture model. Following the detachment of GRP 78, PERK is stimulated, and then it phosphorylates a subunit of eIF2a [80]. The factor eIF2a stops general cap-dependent translation, thus decreasing the further accumulation of proteins within the ER lumen [81]. PERK−/− mouse embryonic fibroblasts lack this translational block and thus are oversensitive to ER stress [81]. This translational block is not absolute, and it does not apply to certain proteins such as ATF4. Following its translation, ATF4 translocates into the nucleus, where it initiates transcription of ER genes and gene products (including GRP 78) involved in amino acid biosynthesis, redox reactions, and protein secretion as well as pro-apoptotic mechanisms including synthesis of the transcription factor CHOP [80]. The downstream ER stress protein ATF4 is a component of the PERK pathway. Measurement of expression levels in this protein serves as an indicator for the extent of the PERK pathway’s response in the presence or in the absence of Carb treatment. We found that there was a marked increase in ATF4 expression, indicating that the PERK pathway is activated in the hypoxia/reoxygenation model of stroke. Our data showed that the PERK pathway is clearly inhibited following the use of Carb in hypoxia/reoxygenation. Upon sensing of ER stress and detachment of GRP 78, ATF6 translocates into the Golgi apparatus. The ratio of cleaved ATF6 to full-length ATF6 demonstrates that Carbamathione increases ATF6 cleavage in PC-12 cells in hypoxia/reoxygenation using the in vitro model of stroke. Certain previous studies have shown protective effects with induced ATF6 cleavage signaling to the UPR [82]. Knockdown of ATF6 in cardiac myocytes exposed to ischemia/reperfusion increased reactive oxygen species and necrotic cell death, both of which were diminished by ATF6 overexpression [83]. Through binding to GRP 78, IRE1 is held in an inactivated state, and upon detachment of GRP 78, IRE1 is activated by dimerization and autophosphorylation. Following its activation, IRE1 allows translation and generation of XBP1 [80]. XBP1 triggers the transcription of many proteins involved in the maintenance of ER homeostasis such as the ER chaperone GRP 78, ER-associated degradation (ERAD) pathway components, as well as transcription factors such as CHOP and XBP1 [84]. XBP-1 showed a significant decrease in response to Carb-treatment in PC-12 cells under hypoxia/reoxygenation. Activation of IRE1-mediated XBP 1 controls absolute levels of CHOP, which has binding sites for ATF6, ATF4, and XBP1s present within its promoter, and CHOP is known to facilitate ER stress-induced cell death through the regulation of Bcl2 family members. Simultaneously, these results indicate that the ER stress response signals to the mitochondrion via the regulation of Bcl2 proteins by CHOP. Our data revealed that CHOP and Bax were up-regulated in the neurons of the BCAO model and showed significantly decreased expression in the brains of the Carb-treated group. We demonstrated that Carb can significantly reduce the expression of CHOP/GADD153 in the PC-12 cells of the Carb-treated group, thus protecting against glutamate excitotoxicity and hypoxia/reoxygenation. These findings provide evidence that activation of the PERK and IRE-1 pathways can be inhibited by Carb, and through these two pathways, PERK may inhibit ER-induced apoptosis. Furthermore, the results indicating the suppression of both CHOP and Bax with Carb treatment provide substantial evidence that Carb can contribute to effective inhibition of the ER stress induced in the in vitro stroke model.

While the ATF-6 pathway showed activation by Carb, the combined inhibitory effects of Carb acting on the IRE-1 pathway as well as on the PERK pathway appear to dominate over the influence of the ATF-6 pathway induction. Clear evidence of the inhibition of the ER stress pathway by Carb is also seen in the decrease of Grp78 and of CHOP following Carb treatment. In addition, pro-survival responses involving Bcl-2 induction and Bax inhibition combined with the effect of Carb on the inhibition of autophagy further support the protective role of Carb.

Caspase-12, which is also activated by the ER stress pathway [9], plays an essential role in programmed cell death progression during the pro-apoptotic phase of the ER stress response [85]. The first ER-associated member of the caspase family is caspase-12, and this novel caspase is associated with the cell death-inducing mechanisms of ER stress [15]. A specific ER membrane-associated caspase-12 is digested to cleaved caspase-12, which induces the caspase pathway cascade [86]. Both the expression of CHOP and the activation of caspase-12 initiate cell death. GRP 78 and caspase-12 are also up-regulated after exposure to hypoxia/reoxygenation. We analyzed the expression of caspase-12 in the absence or presence of Carb after treatment with hypoxia/reoxygenation and demonstrated that caspase-12 or cleaved caspase-12 expression was clearly reduced by Carb following hypoxia/reoxygenation. These results indicating suppression of both CHOP and caspase-12 with Carb treatment provide strong supportive evidence that Carb can contribute to effective inhibition of the ER stress induced by hypoxia/reoxygenation.

In our in vivo study, we provided evidence that Carb reduced infarct volume, improved behavioral scores, and protected against apoptosis by modulating the expression of Bcl2 family members via up-regulating levels of anti-apoptotic Bcl2 and down-regulating proapoptotic Bax expression level. In addition, we demonstrated that Carb was protective against ER stress apoptosis by down-regulating ER stress-induced apoptotic CHOP and down-regulating the expression levels of GRP 78 (ER stress marker).

### Study Limitations and Future Work

Interpretations concerning the potential therapeutic application of Carb are limited by the nature of the current preclinical studies that were carried out using cell culture and mouse models. Using the BCAO model of global ischemia, the current study demonstrated a decreased infarction as well as activation of multiple protective signaling pathways. Demonstration of the clinical relevance of the findings may be dependent upon carrying out further preclinical studies on rodent models, and future large animal studies in addition to potential clinical trials with human subjects. Our previous studies on DETC-MeSO showed protection in a rodent model of focal ischemia 24 h after reperfusion and hence demonstrated a therapeutic time frame that could potentially be very applicable to achieving clinical efficacy. Because BCAO is a more severe insult than focal ischemia, it is anticipated that the protection provided by Carb before occlusion or 30 min after occlusion may also have the potential to be extended to 24 h post-ischemia, as was previously shown for DETC-MeSO. Our future studies will be extended to test Carb in global cerebral ischemia 24 h post-reperfusion. A further limitation of the current study is that the investigation was carried out on male mice. Our previous study on focal ischemia with DETC-MeSO was also conducted on male mice. Since male and female rodents have differences in some aspects of physiological functions including brain functions, presumably due to different actions of male hormones and female hormones, it is imperative to include both male and female animals in future studies to determine the efficacy of Carb as a potential therapeutic agent for stroke.

## 5. Conclusions

The current study demonstrates the potential for the clinical application of Carb as a stroke treatment. Several studies on partial NMDA receptor antagonists show extremely good promise for clinical application by preventing glutamate-induced excitotoxicity while still maintaining effective glutamate neurotransmission. Carb is an excellent candidate for preventing ischemic brain damage because of its pro-survival mechanisms of action (Figure 11) and because it is a metabolite of disulfiram, which has been shown to be safe clinically for the treatment of alcohol abuse.

## Figures and Tables

**Figure 1 biomedicines-11-01885-f001:**
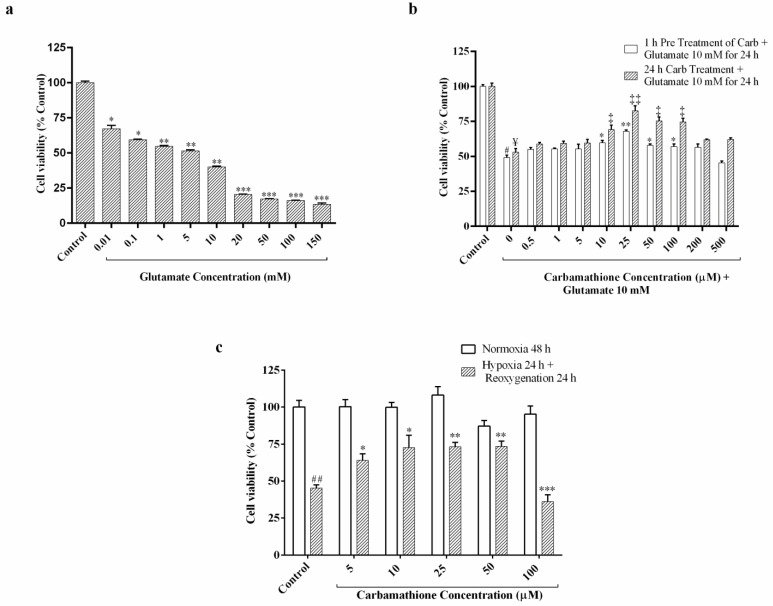
Effect of Carbamathione on glutamate- and hypoxia/reoxygenation-induced cell injury in PC12 cells measured using an ATP assay. (**a**) Dose-dependence of glutamate-induced cell injury in PC-12 cells. Undifferentiated PC-12 cells were exposed to a glutamate concentration range from 0.01 to 150 mM for 24 h. (All data are presented as mean ± SEM (*n* = 6), where * *p* < 0.05, ** *p <* 0.01, *** *p <* 0.001. Asterisks (*) indicate significance relative to the control group). (**b**) Effect of Carbamathione on glutamate (10 mM)-induced cell injury in PC-12 cells. Cell viability was measured using an ATP assay. Briefly, 0.5 µM to 500 µM Carbamathione was pre-incubated for 1 h followed by 10 mM glutamate treatment for 24 h, and 0.5 µM to 500 µM Carbamathione was incubated for 24 h with 10 mM glutamate treatment. (All data are presented as mean ± SEM (*n* = 6), where #/*/‡/¥ *p* < 0.05, ‡‡/** *p <* 0.01, */‡ indicates significant with glutamate 10 mM group, and #/¥ indicate statistical significance relative to the control group). (**c**) Protective effects of Carbamathione on PC-12 cell culture under the hypoxia/reoxygenation condition. Cell cultures were exposed to 0.3% oxygen for 24 h followed by reoxygenation for 24 h. In hypoxia + 5–100 µM Carbamathione, cells were pre-incubated with Carbamathione (5–100 μM) for 1 h before hypoxia. Cell viability was measured using an ATP assay. Normoxia values were fixed at 100% (*n* = 6; * significant compared to the hypoxia group, * *p* < 0.05, ##/** *p <* 0.01, *** *p <* 0.001. Asterisks (*) indicate significance compared to the hypoxia/reoxygenation group. Pound signs (#) indicate statistical significance compared to the control group (Normoxia)).

**Figure 2 biomedicines-11-01885-f002:**
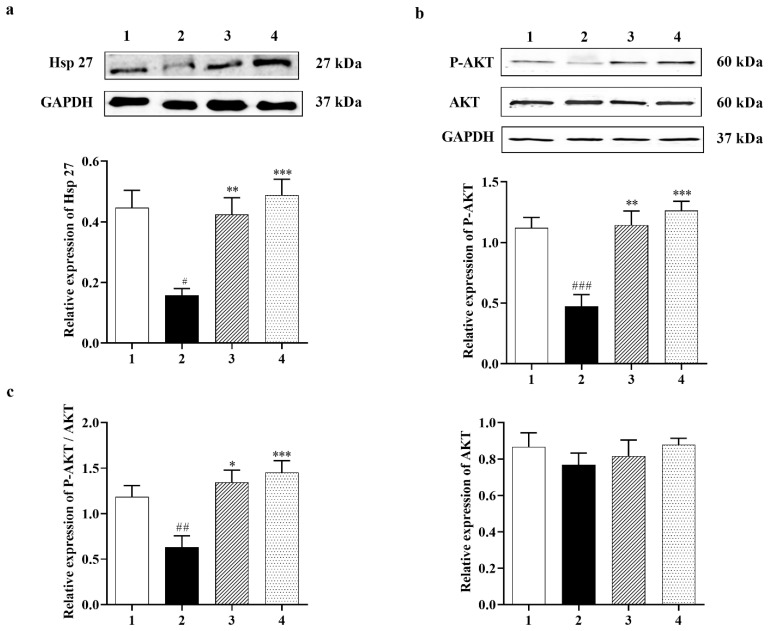
Effect of Carbamathione on the expression of Hsp 27, P-AKT, and AKT after glutamate (10 mM) treatment. Western blot analysis using antibodies against Hsp 27 (**a**) and P-AKT, AKT (**b**). The bar graphs reflect the densitometry data from the respective experiments for Hsp 27 (*n* = 6) (**a**), P-AKT, AKT (**b**), and P-AKT/AKT (**c**) (*F* (3, 14) = 9.226; *p =* 0.0013 (*n* = 5)) Western blot. Note: 1—control, 2—glutamate (10 mM for 24 h), 3—Carbamathione (25 µM for 1 h) followed by glutamate (10 mM for 24 h), and 4—Carbamathione (25 µM for 24 h) with glutamate (10 mM for 24 h). (All data are presented as mean ± SEM, where #/* *p* < 0.05, ##/** *p <* 0.01, ###/*** *p* < 0.001. Pound signs (#) indicate statistical significance between groups 1 and 2; asterisks (*) indicate statistical significance between groups 2 and 3 and between groups 2 and 4).

**Figure 3 biomedicines-11-01885-f003:**
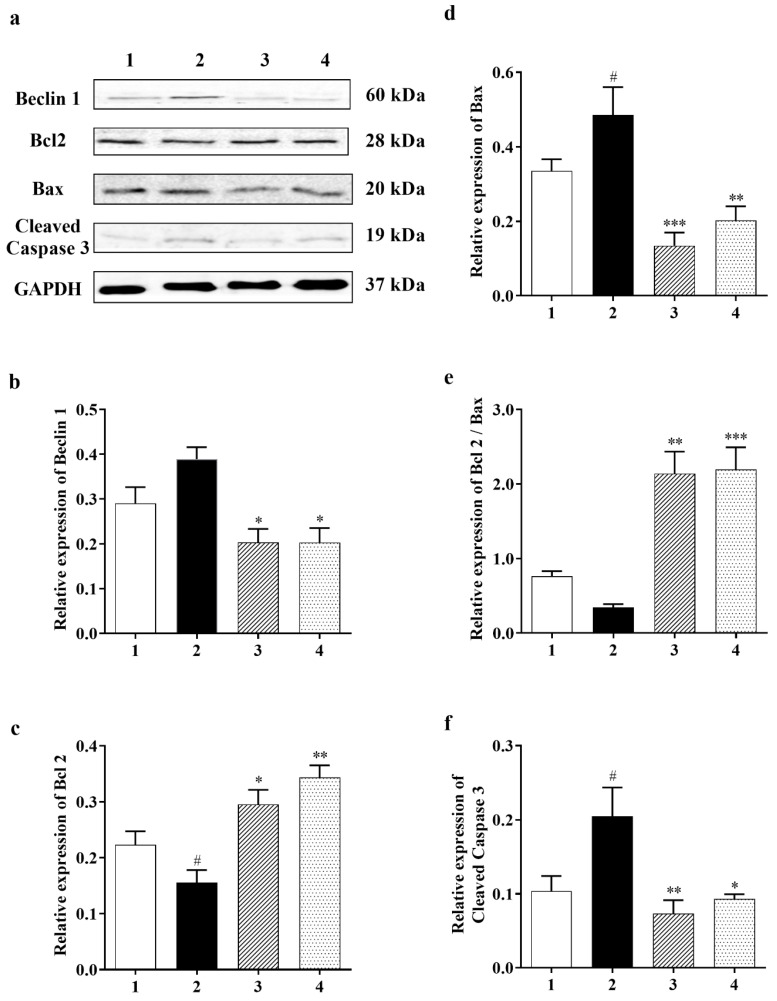
Effect of Carbamathione on the glutamate-induced elevated expression of Beclin 1, Bax, and cleaved caspase-3 and the glutamate-induced decreased expression of Bcl2. (**a**) Western blot analysis using antibodies against Beclin 1, Bcl2, Bax, and cleaved caspase-3. The bar graphs reflected the densitometry data from the respective experiments for Beclin 1 ((**b**) (*n* = 6)), Bcl2 (**c**), Bax (**d**), Bcl2/Bax (**e**), and cleaved caspase-3 (**f**) (*n* = 5) Western blot. Note: 1—control, 2—glutamate (10 mM for 24 h), 3—Carbamathione (25 µM for 1 h) followed by glutamate (10 mM for 24 h), and 4—Carbamathione (25 µM for 24 h) with glutamate (10 mM for 24 h). (All data are presented as mean ± SEM, where #/* *p* < 0.05, ** *p* < 0.01, and *** *p* < 0.001. Pound signs (#) indicate statistical significance between groups 1 and 2; asterisks (*) indicate statistical significance between groups 2 and 3 and between groups 2 and 4).

**Figure 4 biomedicines-11-01885-f004:**
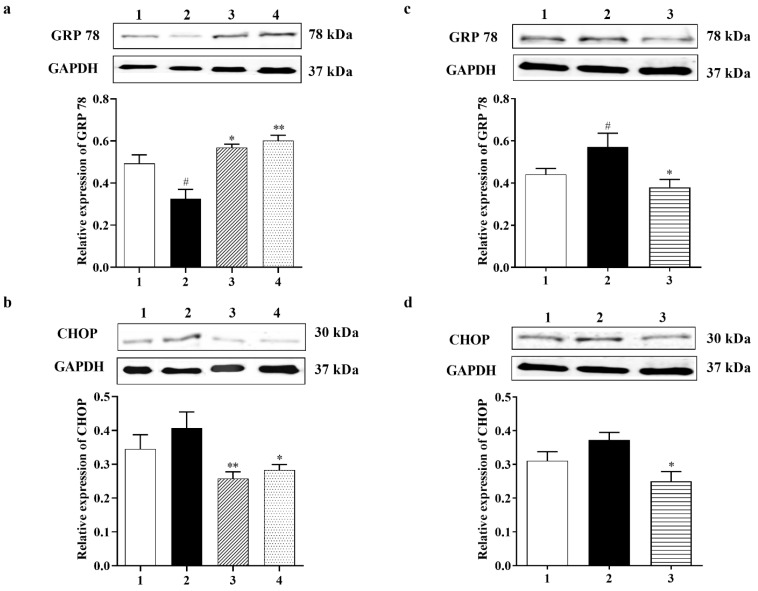
Effect of Carbamathione on ER stress induced by glutamate and hypoxia/reoxygenation. Western blot analysis using antibodies against GRP 78 (**a**,**c**) and CHOP (**b**,**d**).The bar graphs reflect the densitometry data from the respective experiments for GRP 78 (**a**) and CHOP (**b**) Western blot in the glutamate-treated PC-12 cells. Note: 1—control, 2—glutamate (10 mM for 24 h), 3-carbamathione (25 µM for 1 h) followed by glutamate (10 mM for 24 h), and 4—Carbamathione (25 µM for 24 h) with glutamate (10 mM for 24 h). (All data are presented as mean ± SEM (*n* = 5), where #/* *p <* 0.05, ** *p <* 0.01. Pound signs (#) indicate statistical significance between groups 1 and 2; asterisks (*) indicate statistical significance between groups 2 and 3 and between 2 and 4). The bar graphs reflect the densitometry data from the experiment for GRP 78 ((**c**) (*n* = 7)) and CHOP ((**d**) (*n* = 5)) Western blot in the hypoxia/reoxygenation-treated PC-12 cells. Hypoxia (2) = hypoxia (0.3% O_2_) for 24 h, reoxygenation for 24 h; Carb + hypoxia: cells were treated with 25 µM Carbamathione for 1 h, then hypoxia for 24 h, and reoxygenation for 24 h. Note: 1—control (Normoxia 48 h); 2—hypoxia/reoxygenation (48 h); and 3—hypoxia/reoxygenation (48 h) after 1 h, 25 µM Carbamathione preincubation. (All data are presented as mean ± SEM; *#/* p* < 0.05,** *p <* 0.01. Pound signs (#) indicate statistical significance between groups 1 and 2; asterisks (*) indicate statistical significance between groups 2 and 3).

**Figure 5 biomedicines-11-01885-f005:**
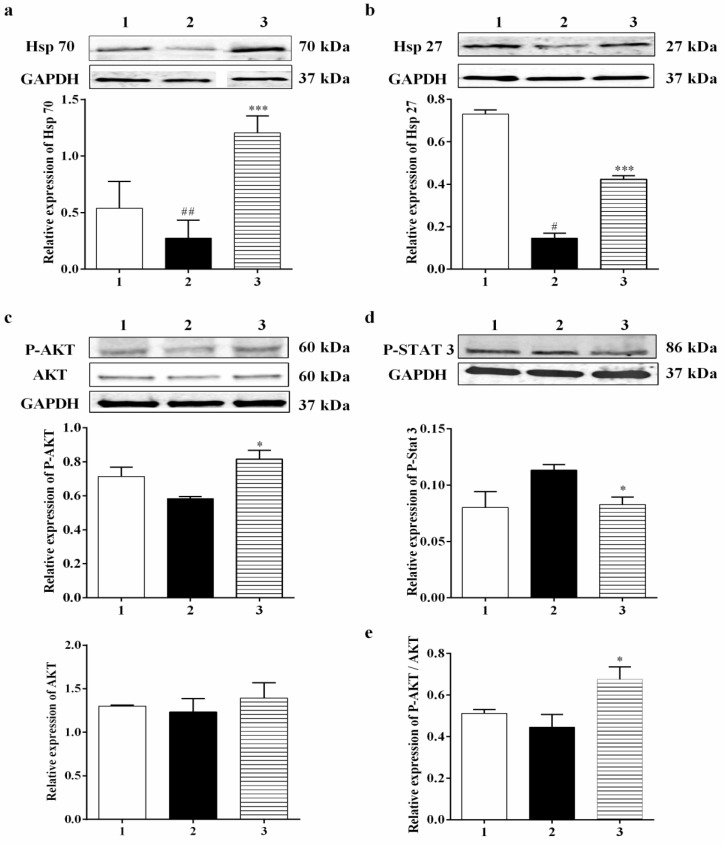
Effect of Carbamathione on the expression of Hsp 70, Hsp 27, P-AKT, AKT, and P-STAT 3 after hypoxia/reoxygenation treatment. Western blot analysis using antibodies against Hsp 70 (**a**), Hsp 27 (**b**), P-AKT, AKT (**c**), and P-STAT 3 (**d**). The bar graphs reflect the densitometry data from the respective experiments for Hsp 70, Hsp 27 (**a**,**b**), P-AKT, AKT (**c**), P-AKT/AKT ((**e**) *(F* (2, 7) = 5.097; *p =* 0.0430 (*n* = 5)), and P-STAT 3 (**d**) Western blots. Hypoxia (Column 2) = hypoxia (0.3% O_2_) for 24 h, reoxygenation for 24 h; Carb + hypoxia: cells were treated with 25 µM Carbamathione for 1 h, then hypoxia for 24 h, and reoxygenation for 24 h. Note: 1—control (Normoxia 48 h); 2—hypoxia/reoxygenation (48 h); and 3—hypoxia/reoxygenation (48 h) after 1 h, 25 µM Carbamathione preincubation. (All data are presented as mean ± SEM (*n* = 5); #/* *p < 0.05*, ## *p <* 0.01, and *** *p <* 0.001. Pound signs (#) indicate statistical significance between groups 1 and 2; asterisks (*) indicate statistical significance between groups 2 and 3).

**Figure 6 biomedicines-11-01885-f006:**
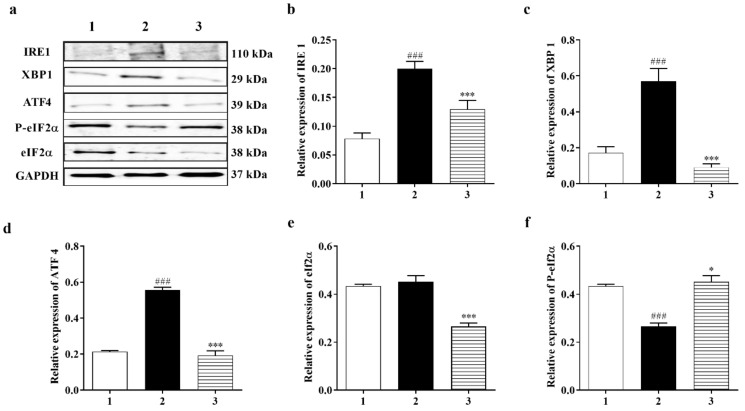
Carbamathione effect on ER Stress pathways after hypoxia/reoxygenation. (**a**) Western blot analysis using antibodies against IRE 1, XBP 1, ATF4, P-eIF2α, and eIF2α. The bar graphs represent the densitometry data from the respective experiments for IRE 1 (**b**), XBP 1 (**c**), ATF4 (**d**), P-eIF2α (**e**), and eIF2α (**f**) Western blots. (**g**) ATF6 and cleaved ATF6 expression analyzed using a Western blot. The bar graphs represent the densitometry data from the respective experiments for ATF6 (**g**), cleaved ATF6 (**h**), and cleaved ATF6/ATF6 (**i**) Western blots. (**j**) Caspase-12 and cleaved caspase-12 expression analyzed using a Western blot. The bar graphs represent the densitometry data from the respective experiments for caspase-12 (**j**) (*n* = 6), cleaved caspase-12 (**k**) (*n* = 7), and cleaved caspase-12/caspase-12 (**l**) *(F* (2, 13) = 5.357; *p* = 0.020 (*n* = 6)) Western blots. Hypoxia (2) = hypoxia (0.3% O_2_) for 24 h, reoxygenation for 24 h; Carb + hypoxia: cells were treated with 25 µM Carbamathione for 1 h, then hypoxia for 24 h, and reoxygenation for 24 h. Note: 1—control (Normoxia 48 h); 2—hypoxia/reoxygenation (48 h); and 3—hypoxia/reoxygenation (48 h) after 1 h, 25 µM Carbamathione preincubation. (All data are presented as mean ± SEM (*n* = 5); #/* *p* < 0.05, ##/** *p* < 0.01, and ###/*** *p <* 0.001. Pound signs (#) indicate statistical significance between groups 1 and 2; asterisks (*) indicate statistical significance between groups 2 and 3).

**Figure 7 biomedicines-11-01885-f007:**
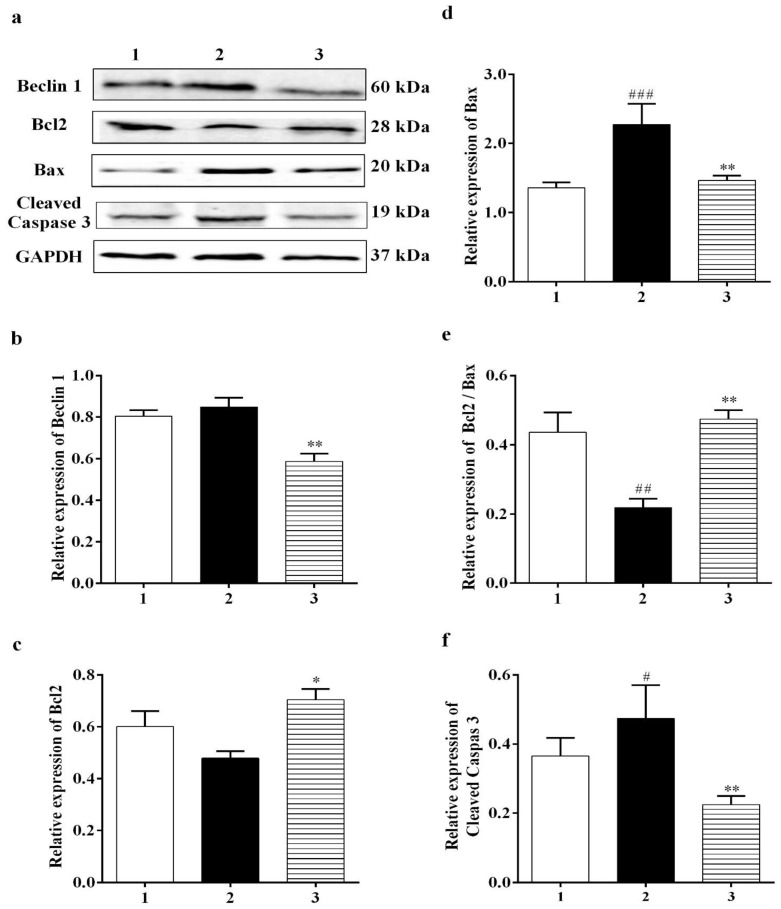
Effects of Carbamathione on hypoxia/reoxygenation-induced changes in Beclin 1 and the Bcl2/Bax ratio. (**a**) Western blot analysis using antibodies against Beclin 1, Bcl2, Bax, and cleaved caspase-3. The bar graphs reflect the densitometry data from the respective experiments for Beclin 1 (**b**), Bcl2 (**c**), Bax (**d**), Bcl2/Bax (**e**), and cleaved caspase-3 (**f**) Western blots. Hypoxia (2) = hypoxia (0.3% O_2_) for 24 h, reoxygenation for 24 h; Carb + hypoxia: cells were treated with 25 µM Carbamathione for 1 h, then hypoxia for 24 h, Note: 1—control (Normoxia 48 h); 2—hypoxia/reoxygenation (48 h); and 3—hypoxia/reoxygenation (48 h) after 1 h, 25 µM Carbamathione preincubation. (All data are presented as mean ± SEM (*n* = 5); #/* *p <* 0.05, ##/** *p* < 0.01, and ### *p <* 0.001. Pound signs (#) indicate statistical significance between groups 1 and 2; asterisks (*) indicate statistical significance between groups 2 and 3).

**Figure 8 biomedicines-11-01885-f008:**
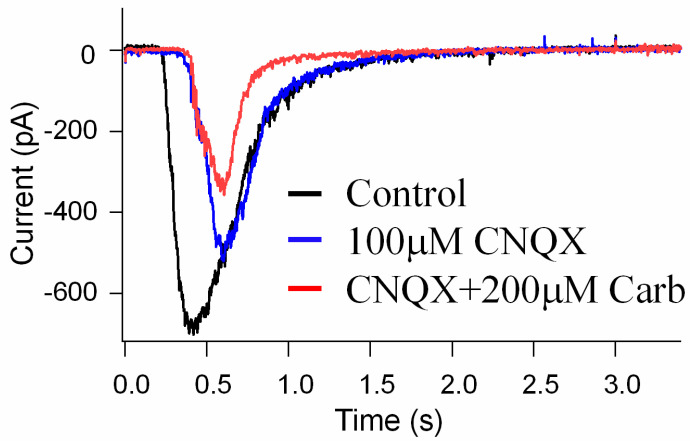
The effects of Carb on the suppression of NMDA glutamate receptor activation in retinal neurons. Light-evoked glutamate currents were recorded in retinal ganglion cells in whole-cell patch-clamp recordings. CNQX blocks synaptic non-NMDA receptors (blue trace), revealing NMDA receptor currents; Carb inhibits NMDA receptor currents (red trace) in the ganglion cells (*n* = 3).

**Figure 9 biomedicines-11-01885-f009:**
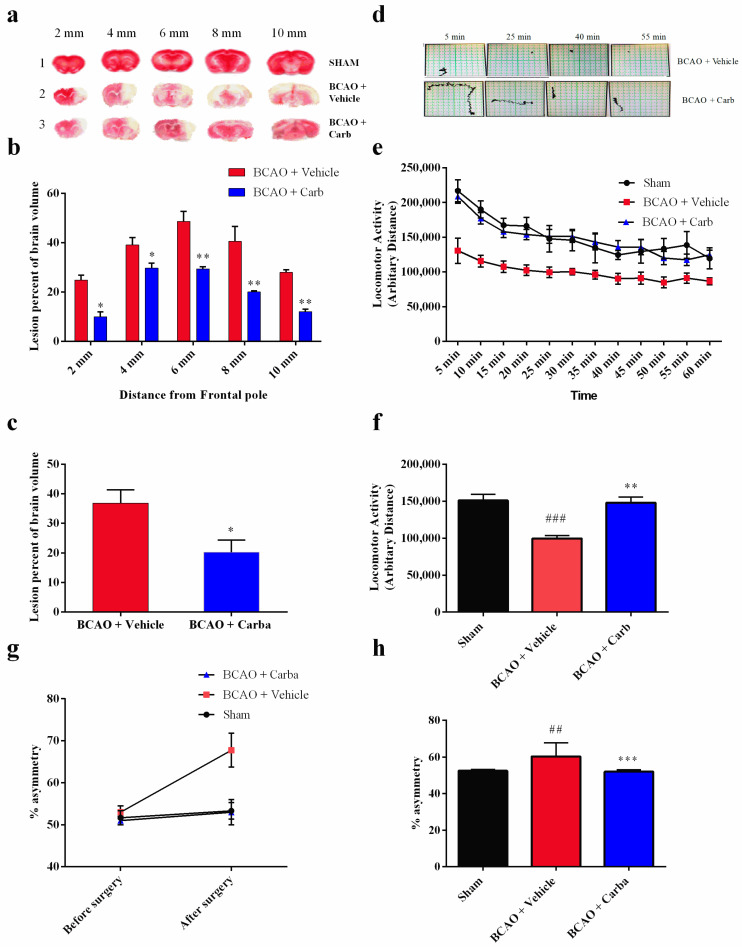
Morphological analysis of the effect of Carbamathione on ischemia-induced brain injury in the BCAO stroke model and the effect of Carbamathione on the corner test and locomotor activity test after BCAO. (**a**) The TTC brain slices 2, 4, 6, 8, and 10 mm from the frontal pole of the sham group, vehicle-treated group, and Carbamathione-treated group 30 min after occlusion at 4 days. Note: 1—sham group, 2—vehicle-treated group, and 3—Carbamathione-treated group 30 min after occlusion. (**b**) The TTC results for the infarct size of the brain slices from the vehicle-treated group and Carbamathione-treated group 4 days after 30 min occlusion. The quantitative analysis revealed that both treated groups after 4 days produced a significant reduction in the infarction percentage. The sham-operated group showed no infarct zone. (**c**) The TTC results for the whole-brain infarct size in the vehicle-treated group and Carbamathione-treated group 4 days after 30 min occlusion. Data represent infarct volume as a percent of both hemisphere volumes, and the values are mean ± SEM of 9 experiments for BCAO plus vehicle and BCAO plus Carbamathione at 4 days. (*n* = 9, * *p* < 0.05, ** *p* < 0.01). (**d**) Sample traces. (**e**,**f**) Locomotor summary. The locomotor activity of BCAO mice with and without Carbamathione treatment was measured after 4 days. Results show an increased amount of activity with the administration of Carbamathione. The summary of locomotor data is provided in (**e**) and expressed as mean ± SEM, where *n* = 9 for sham, BCAO/vehicle, and BCAO/Carb. The data show that animals treated with Carbamathione have activity consistent with the sham animals. (**f**) Average results (*n* = 9) for the locomotor test densitometric scanning are presented. (**g**) Percentage asymmetry is observed as a mouse enters a 30° corner and either turns left or right. In the above graph (**g**), percent asymmetry is compared before and after BCAO for sham, BCAO/vehicle, and BCAO/Carbamathione (*n* = 9). It is observed that mice treated with Carbamathione after BCAO had a significantly better percent asymmetry than those treated with BCAO/vehicle and similar to the percent asymmetry of the sham mice. (**h**) Average results (*n* = 9) for the corner test densitometric scanning are presented. (All data are presented as mean ± SEM; * *p* < 0.05, ##/** *p* < 0.01, and ###/*** *p* < 0.001. For (**c**,**f**,**h**), the pound signs (#) show statistical significance between the sham and BCAO + vehicle groups; for (**c**,**f**,**h**), the asterisks (*) show statistical significance between the BCAO + vehicle and BCAO + Carb groups).

**Figure 10 biomedicines-11-01885-f010:**
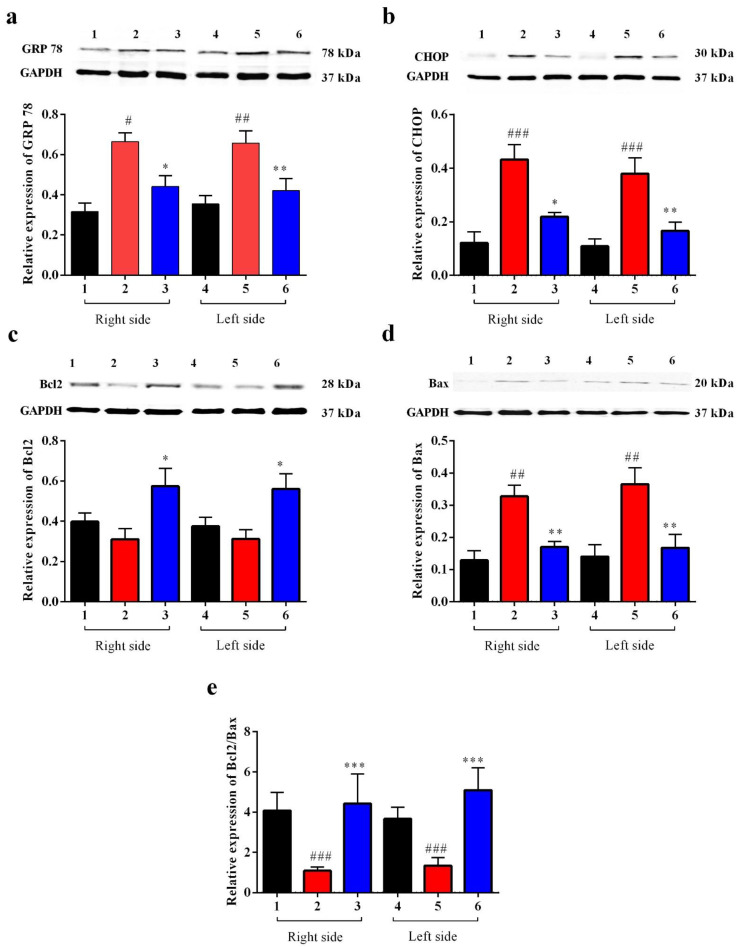
Effect of Carbamathione on the expression of UPR (GRP 78), the CHOP protein, and the Bcl2/Bax ratio in the BCAO stroke model. (**a**,**b**) Results from the Western blot analysis as described in the Materials and Methods section. (**a**) Average results (*n* = 9) for GRP 78 densitometric scanning are presented. (**b**) Average results (*n* = 9) for CHOP densitometric scanning are presented. (**a**,**b**) Results from the Western blot analysis as described in the Materials and Methods section. (**c**) Average results (*n* = 9) for Bcl2 densitometric scanning are presented. (**d**) Average results (*n* = 9) for Bax densitometric scanning are presented. (**e**) The graph shows the ratio of Bcl2 to Bax in BCAO brains in the Carbamathione-treated group and the vehicle-treated group. Note: 1,4—sham; 2,5—BCAO with vehicle treatment; 3,6—BCAO with Carbamathione treatment; 1,2,3—right side of the mouse brain; and 4,5,6—left side of the mouse brain. (All data are presented as mean ± SEM, where #/* *p* < 0.05, ##/** *p* < 0.01, and ###/*** *p* < 0.001. Pound signs (#) show statistical significance between groups 1 and 2 and between groups 4 and 5; asterisks (*) show statistical significance between groups 2 and 3 and between 5 and 6).

**Figure 11 biomedicines-11-01885-f011:**
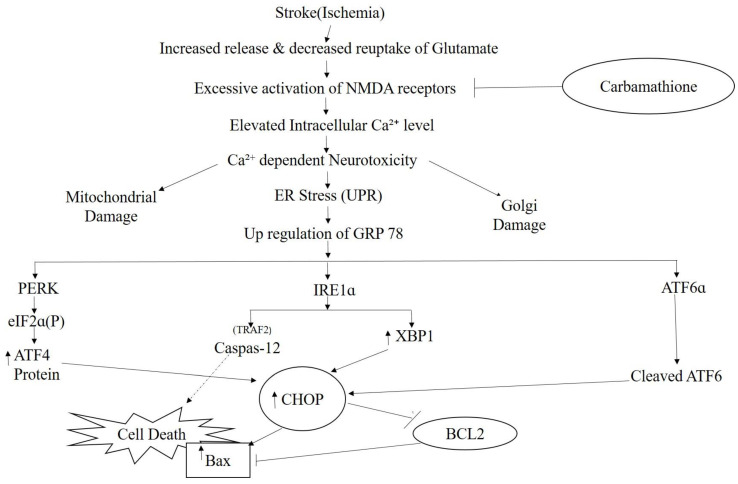
Scheme showing the protective effects of Carbamathione against the activation of ER stress pathways. After neurons are subjected to glutamate or hypoxia, the homeostasis in neuronal culture is disturbed, which initiates dimerization and autophosphorylation of the ER membrane proteins PERK and IRE1. ATF6 (P90) is activated by limited proteolysis after its translocation from the ER into the Golgi apparatus to form cleaved ATF6 (P50). Activated PERK phosphorylates eIF2a, which induces ATF4 expression. ATF4, being a transcription factor, translocates into the nucleus and induces the transcription of genes required to block the translational pathway. These three pathways will induce the up-regulation of CHOP. Caspase-12, a specific ER membrane-associated caspase, is proteolyzed to cleaved caspase-12, which induces the caspase pathway cascade. Both the expression of CHOP and the activation of caspase-12 initiate cell death. Carbamathione treatment greatly inhibits the PERK and IRE1 pathways but does not inhibit the ATF6 pathway after hypoxia/reoxygenation.

## Data Availability

Not applicable; no data sets appropriate for depositing in a repository were generated in this study. The drug Carbamathione is a small-molecule metabolite of disulphiram and can be generated by investigators when needed. Therefore, it is generally available to the scientific community. The datasets used and/or analyzed during the current study are available from the corresponding author upon reasonable request.

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
