# Peer review of "The Role of NMDA Receptor Partial Antagonist, Carbamathione, as a Therapeutic Agent for Transient Global Ischemia"

_biomedicines, 2023, doi:10.3390/biomedicines11071885_

Round 1
Reviewer 1 Report
This study evaluated the role carbamathione as a therapeutic agent in animal models of transient global ischemia.
The originality of the work and the comprehensive experimental approach are the main points of strengths in my view.The study is interesting, well performed and well described. There are, however, some issues that should be further addressed.
It would be fine to add a few comments about if and how this approach may be translated to humans, and which could be the expected results.
Minor edits required.
Author Response
Thanks for the positive comments. The response to the comments “about if and how this approach may be translated to humans, and which could be the expected results” is as follows:
Previous clinical studies using glutamate receptor antagonists (such as MK801) for stroke were unsuccessful. They were reported to have failed because of the complete blocking of the glutamate neurotransmission system. Carb is an excellent candidate for neuroprotection against ischemia-induced brain damage through preventing glutamate induced excitotoxity while still maintaining effective glutamate neurotransmission. Furthermore, Carb is safe because it is a metabolite of disulfiram which has been shown to be safe clinically in the treatment of alcohol abuse for six decades. Hence, Carb has a great potential for clinical application for stroke treatment.
This additional comment is now included in the Discussion section, lines 1045-1053
Comment: Reviewer 1 pointed out that there were minor grammatical errors.
Response: The manuscript has now been further edited to remove grammatical errors and the changes are highlighted in the manuscript.
Reviewer 2 Report
Carbamathione (Carb), a partial antagonist of the NMDA glutamate receptor, exhibits potent neuroprotective effects against neuronal injury caused by hypoxia or ischemia in stroke models conducted using cells or animals. We utilized PC-12 cell cultures as a cellular model and employed bilateral carotid artery occlusion (BCAO) to induce stroke. In this work the authors conducted whole-cell patch clamp recordings in mouse retinal ganglion cells. Immunoblotting was performed to analyze key proteins involved in apoptosis, endoplasmic reticulum (ER) stress, and heat shock proteins. Carb demonstrates effectiveness in safeguarding PC12 cells against injury induced by glutamate or hypoxia. Electrophysiological results reveal that Carb diminishes NMDA-mediated glutamate currents in retinal ganglion cells, leading to the activation of the AKT signaling pathway and an increase in the expression of pro-cell survival biomarkers such as Hsp 27, P-AKT, and Bcl2. Furthermore, it decreases the expression of pro-cell death markers including Beclin 1, Bax, Cleaved caspase 3, as well as ER stress markers such as CHOP, IRE1, XBP1, ATF 4, and eIF2α. In the BCAO animal stroke model, Carb reduces brain infarct volume and lowers the levels of ER stress markers GRP 78 and CHOP. At the behavioral level, it results in a decrease in asymmetric turns and an increase in locomotor activity. These findings highlight the promising and rational strategies offered by Carb for stroke therapy.
Comments: This is a nice technical flawless study. However, we have difficulties with the translation of experimental studies done on animals to humans. Then, what we say about the in vitro studies? Are they really necessary? Another problem is the use of young animals for a condition that usually affects older individuals (see, doi: 10.1155/2019/9785476)
English is fine
Author Response
Thanks for the positive comments. The response to the comments “However, we have difficulties with the translation of experimental studies done on animals to humans. Then, what we say about the in vitro studies? Are they really necessary? Another problem is the use of young animals for a condition that usually affects older individuals” is as follows:
The rationale to include a cell-based stroke model is to analyze the mechanism of neuroprotective function of Carb in cultured cells, a much simpler system compared to an animal system, and therefore, allows one to obtain more definitive information at molecular level. In addition, it is always desirable to show the proof-of -concept using two different systems (Please see the Discussion section, lines 785-796). The rationale to use young mice in the current study is to avoid complications associated with aged animals. The reviewer’s comment that stroke usually affects older individuals is highly relevant and is definitely desirable to be included in the future study.
This addition is added in the Materials and Methods Section, lines 182 -184.
Comment: Reviewer 2 pointed out that there were minor grammatical errors.
Response: The manuscript has now been further edited to remove grammatical errors and the changes are highlighted in the manuscript.
Round 2
Reviewer 2 Report
The authors have addressed my concerns
Author Response
Thanks For the Positive comments. Please see attachment